# Dual functions of microRNA-17 in maintaining cartilage homeostasis and protection against osteoarthritis

Yun Zhang [1], Shuaijun Li[2], Peisheng Jin[3], Ting Shang[4], Ruizhu Sun[2], Laiya Lu[1], Kaijin Guo[5], Jiping Liu[1], Yongjuan Tong[4], Junbang Wang[6], Sanhong Liu[7], Chen Wang [7], Yubin Kang[4], Wenmin Zhu[1], Qian Wang[2], Xiaoren Zhang[8], Feng Yin[1], Yi Eve Sun[1✉] & Lei Cui [1,2,4✉]

Damaged hyaline cartilage has no capacity for self-healing, making osteoarthritis (OA) "difficult-to-treat". Cartilage destruction is central to OA patho-etiology and is mediated by matrix degrading enzymes. Here we report decreased expression of miR-17 in osteoarthritic chondrocytes and its deficiency contributes to OA progression. Supplementation of exogenous miR-17 or its endogenous induction by growth differentiation factor 5, effectively prevented OA by simultaneously targeting pathological catabolic factors including matrix metallopeptidase-3/13 (MMP3/13), aggrecanase-2 (ADAMTS5), and nitric oxide synthase-2 (NOS2). Single-cell RNA sequencing of hyaline cartilage revealed two distinct superficial chondrocyte populations (C1/C2). C1 expressed physiological catabolic factors including MMP2, and C2 carries synovial features, together with C3 in the middle zone. MiR-17 is highly expressed in both superficial and middle chondrocytes under physiological conditions, and maintains the physiological catabolic and anabolic balance potentially by restricting HIF-1α signaling. Together, this study identified dual functions of miR-17 in maintaining cartilage homeostasis and prevention of OA.

[1] Shanghai Institute of Stem Cell Research and Clinical Translation & Institute for Regenerative Medicine & Department of Joint Surgery, Shanghai East Hospital, Tongji University School of Medicine, Shanghai, China. [2] Key Laboratory of spine and spinal cord injury repair and regeneration, Ministry of Education of the People's Republic of China & Department of Orthopedics, Tongji Hospital, Tongji University School of Medicine, Shanghai, China. [3] Department of Plastic Surgery, Affiliated Hospital of Xuzhou Medical University, Xuzhou, China. [4] Department of Plastic Surgery, Beijing Shijitan Hospital, Capital Medical University, Beijing, China. [5] Department of Orthopedics, Affiliated Hospital of Xuzhou Medical University, Xuzhou, China. [6] Translational Stem Cell Research Center, Tongji Hospital, Tongji University School of Medicine, Shanghai, China. [7] Shanghai Institute for Advanced Immunochemical Studies, ShanghaiTech University, Shanghai, China. [8] Guangzhou Municipal and Guangdong Provincial Key Laboratory of Protein Modification and Degradation, State Key Laboratory of Respiratory Disease, Affiliated Cancer Hospital & Institute of Guangzhou Medical University, Guangzhou, China. ✉email: yi.eve.sun@gmail.com; 14209@tongji.edu.cn

Osteoarthritis (OA) is the most prevalent degenerative disorder of synovial joints and leads to chronic disability due to pain and associated joint dysfunction[1]. Currently, no treatments can effectively impede the progression of OA[2], and surgical joint replacement is ultimately required in many affected patients[3]. The structural integrity of healthy cartilage is maintained by a fine-tuned balance between anabolism and catabolism of extracellular matrix (ECM) components in chondrocytes[4]. Chondrocytes display zone-specific heterogeneity amongst different layers of the articular cartilage[5,6]. However, during OA pathogenesis, the cartilage matrix is gradually degraded owing to excessive expression of multiple matrix-destructive enzymes, particularly matrix metallopeptidases MMP3, MMP13, and aggrecanase-2 (ADAMTS5), produced by chondrocytes in response to biochemical and biomechanical stimuli[7]. Thus, exploring molecular mechanisms underlying the imbalance between anabolic and catabolic pathways is expected to help identify new therapeutic targets for OA.

MicroRNAs (miRNAs) had been reported to modulate various pathological processes during OA by repressing their target genes. Supplementation of those reported individual miRNAs has been shown to elicit detectable levels of protection against OA, yet without robust repair[8–10]. Targets of those reported miRNAs were almost always limited to single catabolic factors, while a promising therapeutic option for OA may require concomitant inhibition of multiple matrix-degrading enzymes. Perhaps a combinatorial usage of multiple miRNAs is an option, unless one microRNA can be identified to simultaneously target multiple, if not all of the key OA-associated matrix-degrading enzymes.

MiR-17-5p (miR-17) belongs to the *miR-17~92* cluster[11], and dysregulation of this microRNA cluster has been found to be associated with skeletal malformation and related growth defects in humans[12]. In a published study by Han et al. through serial mutations of each of the 4 seed family members within this cluster, only deletion of miR-17 seed family resulted in vertebral transformations and shortening of the fifth mesophalanx, indicating that miR-17 has a profound role in controlling skeletal development[13]. However, the function of the *miR-17~92* cluster, especially miR-17, has not been fully elucidated in adult cartilage maintenance and OA progression. The *miR-17~92* cluster could be regulated by BMP family proteins including BMP2/4 during cardiac development[14]. Another BMP family member, growth differentiation factor 5 (GDF-5) plays key roles during joint morphogenesis[15]. Functional single-nucleotide polymorphisms linked to reduced *Gdf5* gene transcription are currently the most consistent risk factor for adult-onset OA across human populations[16,17]. Interestingly, skeletal defects resulted from dysfunction of *miR-17~92* cluster showed shortening of phalangeal elements and bony fusions of the joints[12], which are also observed with GDF5 deficiency[18], suggesting an inter-connection between GDF5 and miR-17.

In this study, we used destabilization of the medial meniscus (DMM)-induced mouse OA model, as well as a series of gain- and loss-of-function analyses plus single-cell RNA sequencing (scRNA-seq) to examine the regulation and function of miR-17, together with molecular mechanisms underlying OA pathophysiology.

## Results

**MiR-17 reduction plays a critical role in OA progression.** We performed in-silico screening using 30 most-abundantly expressed miRNAs identified in chondrocytes[19], as well as miRNA target prediction programs to search for miRNAs predicted to simultaneously target MMP13 and ADAMTS5. Such a screen resulted in 4 miRNAs, which are miR-17, miR-20a, miR-140-3p, and miR-27-3p (Supplementary Fig. 1a). MiR-140-3p and miR-

27-3p have been reported to play essential roles in cartilage development and homeostasis[20,21]. Interestingly, miR-17 and miR-20a happened to be located in the *miR-17~92* cluster. The *miR-17~92* cluster contained a total of 6 microRNA members including the miR-17 family (miR-17 and miR-20a), the miR-18 family (miR-18a), the miR-19 family (miR-19a-3p and miR-19b-3p), and miR-92a-3p. We investigated a possible relationship between GDF-5 and *miR-17~92* cluster, and found that miR-17 was the most highly expressed and strongly upregulated miRNA within this cluster, in response to GDF-5. MiR-20a was also significantly upregulated by GDF-5 (Supplementary Fig. 1b). Since miR-17 shared the same seed sequence with miR-20a and had a higher expression level, we selected miR-17 for further study.

In uninjured cartilage, miR-17 expressing cells were mainly located in the superficial and transitional (middle) zones, accounting for 31.32% and 64.91% respectively, of all miR-17 positive cells in the cartilage as detected by fluorescence in situ hybridization (FISH) (Fig. 1a, Supplementary Fig. 1c). In DMM-induced OA mice, Osteoarthritis Research Society International (OARSI) grade scores increased overtime, and percentages of miR-17-positive cells over total cells in the cartilage declined from 41.99% to 8.03%, by week 8 and on-ward (Fig. 1a, b). At 4 weeks after DMM surgery, the percentages of miR-17-positive cells dropped from 70.39% over total DAPI$^+$ cells within the superficial layer to 30.83%, and from 65.75% to 36.1% in the middle layer (Supplementary Fig. 1d). qRT-PCR also demonstrated consistent decreases of miR-17 expression overtime after surgery (Supplementary Fig. 1e), consistent with previous report[22]. To explore the role of miR-17 reduction in OA progression, we injected antagomir-17 at dose of 3 nmol intra-articularly to neutralize the endogenous miR-17 (Supplementary Fig. 1f). This administration indeed aggravated the cartilage destruction and increased OARSI scores at 4 weeks post DMM surgery (Fig. 1c). In sham-operated mice, however, joints received antagomir-17 injection still manifested normal cartilage architecture (Fig. 1c), suggesting inhibition of miR-17, per se, is not sufficient to cause cartilage damage. For gain-of-function study, injection of 1.5 nmol agomir-17 into DMM-induced joints weekly, starting at 4 weeks post DMM surgery, significantly increased miR-17 expression as well as numbers of miR-17-positive cells, along with improved OARSI scores, after 4 injections (Fig. 1d, Supplementary Fig. 1g, h). In sham-operated mice, agomir-17, like antagomir-17, also had no influence on the cartilage structure (Fig. 1d), suggesting endogenous levels of miR-17 are already suffice and excessive miR-17 does not elicit more protection, yet it does not lead to harmful outcome either.

Genetically, the miR-17 gene is located in the *miR-17~92* polycistronic cluster. We generated conditional knockout (cKO) mice with chondrocyte-specific *miR-17~92* deletion, showing reduction of miR-17 levels (Supplementary Fig. 1i). Immediately following tamoxifen treatment, DMM surgery was performed on both control and cKO mice (8 weeks old). Four weeks later, cKO mice already developed OA with extensive loss of cartilage and higher OARSI scores, whereas control DMM mice had only shown mild cartilage degradation without showing significantly higher OARSI scores compared to non-injury controls (Fig. 1a, e). It was obvious that in DMM joints, deletion of *miR-17~92* cluster resulted in greater severity of cartilage destruction than antagomir-17 injection. qRT-PCR indicated that expression levels of miR-17 in cartilage were substantially higher than those of the other five members, while miR-19b-3p (miR-19) and miR-92-3p (miR-92) were also highly expressed (Supplementary Fig. 1j). In order to investigate the function of individual miRNA within the *miR-17~92* cluster in OA progression, we performed rescue

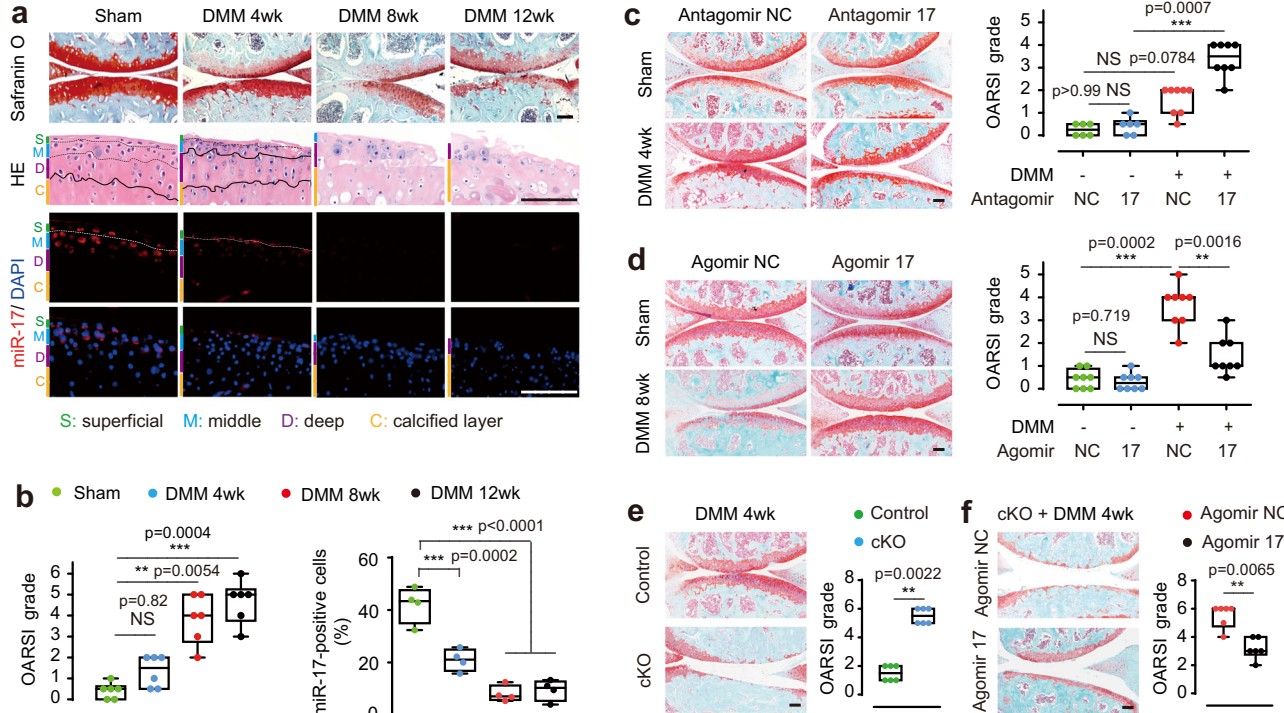

**Fig. 1 MiR-17 deficiency exacerbated cartilage destruction in DMM-induced OA. a** Representative images from the safranin O/fast green and H&E staining, as well as FISH analysis of miR-17 in knee joint sections at 4, 8, or 12 weeks after sham operation or DMM surgery. Male mice underwent surgeries at the age of 10-week old. Cartilage structure was visualized by safranin O/fast green. H&E staining showed layers of articular cartilage. The color bars at the left side of pictures were used to demonstrate the different layers. The blue staining by DAPI showed nuclei to visualize the cellular location of miR-17. The images are representative of three independent experiments. **b** OARSI scores ($n = 6$ mice at each timepoint) and percentage of miR-17-positive cells in FISH staining ($n = 4$ mice at each timepoint) from mice at indicated timepoints after operation. OARSI scores were used to quantify the cartilage destruction from safranin O staining sections. **c** Safranin O staining and scoring of OARSI grade in mice subjected to sham operation or DMM surgery and weekly intra-articular injection of antagomir-NC or antagomir-17 (3 nmol) for three times, starting from 1 week after surgery. $n = 6$ mice (sham + antagomir-NC, sham + antagomir-17); $n = 8$ mice (DMM + antagomir-NC, DMM + antagomir-17). **d** Representative images of safranin O/fast green staining of joint sections and OARSI scores. Agomir-NC or agomir-17 (1.5 nmol) was injected into knee joints 4 weeks after sham operation or DMM surgery. The injections were performed weekly for 4 times. $n = 8$ mice per group. **e** Representative images of safranin O/fast green staining and OARSI grade from *miR-17~92* cKO mice and their control littermates (*miR-17~92^fl/fl*) at 4 weeks after DMM surgery. $n = 6$ mice per group. The cKO mice and their controls were injected with tamoxifen at the age of 7-week old and performed surgeries at the age of 8-week old. **f** Representative images of safranin O/fast green staining and OARSI grade in *miR-17~92* cKO mice subjected to DMM surgery and intra-articular injection of agomir-NC, or agomir-17 (2 nmol). The injections were preformed weekly for 3 times beginning at 1 week after surgery. $n = 6$ mice per group. Data are presented as boxplots. Boxplots: center line, median; box limits, 25 to 75th percentiles; whiskers, min to max. NS not significant, *$P < 0.05$, **$P < 0.01$, and ***$P < 0.001$. One-way ANOVA (nonparametric) with Dunn's multiple comparisons test for OARSI grade in (**b**, **c**); Mann–Whitney $U$ test (two sided) for (**d**, **e**, **f**); one-way ANOVA with Bonferroni's test for percentage of positive cells in (**b**). All scale bars, 100 μm. All the analyses were sourced from tibial cartilage. Source data are provided as a Source Data file.

experiments by injecting each miRNA (miR-17, miR-19 and miR-92) into joints of cKO mice. As expected, supplementation with agomir-17 from 1 week after DMM surgery in cKO mice, rescued cartilage destruction and improved OARSI scores by 4 weeks post-surgery (Fig. 1f). Agomir-19 did not elicit significant cartilage protection, while agomir-92 demonstrated certain degrees of protection, but less than that of agomir-17 (Supplementary Fig. 1k), which further substantiate the role of miR-17 in protection against OA. Aging is a well-known prominent risk factor for OA[23], we analyzed miR-17 expression in 3- and 12-month-old mice and found decreased miR-17 expression with aging (Supplementary Fig. 2a). After DMM surgery, aged (12-month-old) mice showed more extensive cartilage destruction (Supplementary Fig. 2b). Strikingly, agomir-17 was still effective in protecting aged DMM mice (Supplementary Fig. 2c).

**miR-17 suppresses cartilage destruction by inhibiting multiple catabolic factors**. To explore mechanisms by which miR-17

inhibits cartilage destruction during OA progression, we used multiple microRNA target prediction programs. Nine potential targets relevant to catabolism in OA cartilage were selected, including *Epas1* (encoding HIF-2α), *Mmp2*, *Mmp3*, *Mmp13*, *Mmp24*, *Adamts5*, *Adamtsl2*, *Adam9* and *Nos2* (nitric oxide synthase-2), in addition to previously published targets[24,25]. We examined expression patterns of these genes in OA chondrocytes and data indicated that 5 genes (*Epas1*, *Mmp3*, *Mmp13*, *Adamts5*, and *Nos2*) were elevated significantly in IL-1β-treated chondrocytes as well as DMM cartilage (Supplementary Fig. 3a, b). After confirming that *Epas1* was not a direct target for miR-17 (Fig. 2a), we selected 3′-UTRs of *Mmp3*, *Mmp13*, *Adamts5* and *Nos2* for further characterization (Supplementary Fig. 3c). Luciferase reporter assays verified that miR-17 suppressed the expression of all 4 genes through 3′UTR seed recognition sequences, because mutations of which completely eradicated miR-17-induced suppression (Fig. 2a). In cultured mouse chondrocytes, exposure to proinflammatory factor IL-1β resulted in significant upregulation of all 4 catabolic factors (MMP3,

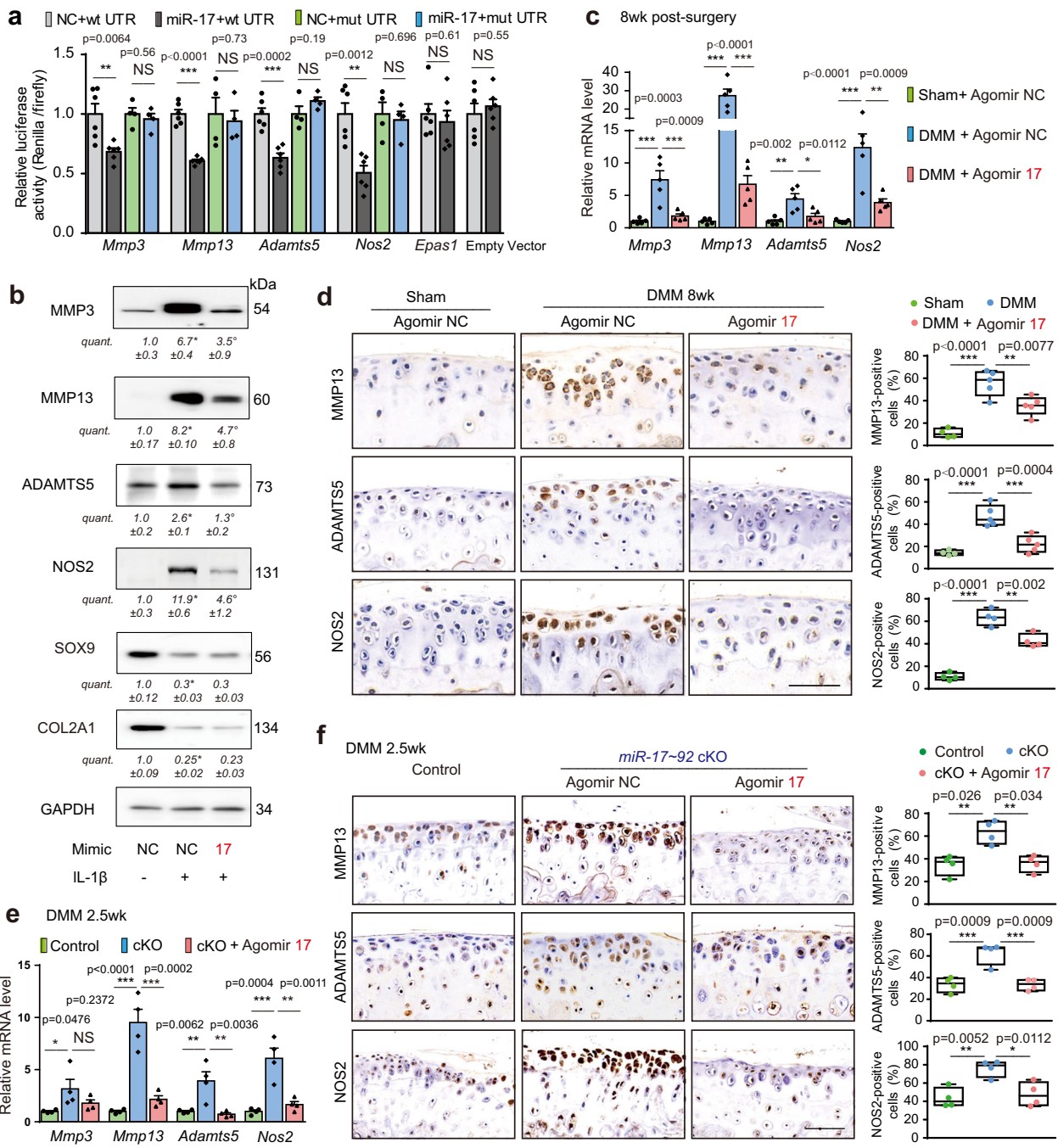

MMP13, ADAMTS5 and NOS2), which could be partially reversed by transfection of miR-17 mimic (Fig. 2b). However, SOX9 and COL2A1, which were suppressed by IL-1β, could not be reversed by miR-17 (Fig. 2b), indicating they are not direct targets for miR-17. In agreement with the in vivo results, miR-19 could only suppress 2 catabolic factors, ADAMTS5 and NOS2, while miR-17 and miR-92 could regulate all 4 catabolic factors (Supplementary Fig. 4). In DMM cartilage, in vivo, application of agomir-17 was found to suppress all 4 catabolic factors at mRNA levels (Fig. 2c). Moreover, numbers of MMP13+, ADAMTS5+ or NOS2+ cells were also decreased upon agomir-17 treatment (Fig. 2d). Similar phenotypes were observed in DMM cKO mice and DMM with antagomir-17 injections (Fig. 2e, f, Supplementary Fig. 5). Together, these findings demonstrated that miR-17 protected against cartilage destruction by targeting multiple pathological catabolic factors in OA pathologies.

**GDF-5 upregulated endogenous miR-17 to protect against OA.**
To explore whether similar levels of protection against OA could be achieved through upregulation of endogenous miR-17, we selected GDF-5 as an inducer for miR-17. In cultured mouse articular chondrocytes, exogenous GDF-5 induced expression of miR-17 under IL-1β stimulation (Supplementary Fig. 6a). Moreover, GDF-5 markedly suppressed IL-1β-induced expression of MMP3, MMP13, ADAMTS5 and NOS2, in a dose-dependent manner, which could be reversed by miR-17 inhibitor, at both mRNA and protein levels (Supplementary Fig. 6b, c). More importantly, intra-articular injection of GDF-5 restored miR-17 expression and numbers of miR-17+ cells in DMM cartilage, along with reversed OARSI scores (Fig. 3a, b, Supplementary Fig. 6d), while intra-articular administration of antagomir-17 together with GDF-5, largely abrogated the protective effect of GDF-5 (Fig. 3c–e). We further showed that repeated GDF-5

**Fig. 2 miR-17 inhibits cartilage destruction by targeting multiple pathological catabolic factors. a** Luciferase activity of 293 T cells co-transfected with dual-luciferase reporter constructs containing wild-type or mutated 3′-UTRs, as well as negative control (mimic NC) or miR-17 mimic. The data presented are mean percentage changes over mimic NC ± s.e.m. $n = 6$ biologically independent samples (NC + wt UTR, miR-17+wt UTR); $n = 4$ biologically independent samples (NC + mut UTR, miR-17 + mut UTR). **b** Mouse articular chondrocytes were transfected with miR-17 mimic (50 nM) or mimic NC and treated with or without IL-1β (5 ng/mL) for 24 h. The protein levels of catabolic and anabolic factors were determined by western blot followed by densitometry analysis. Blots are representative of three independent experiments. **c, d** qRT-PCR analysis of catabolic genes in knee cartilage (**c**), and representative images of IHC staining and quantification of MMP13[+], ADAMTS5[+] and NOS2[+] cells in joint sections (**d**) of mice subjected to sham or DMM surgery and intra-articular injections of agomir-NC or agomir-17 (1.5 nmol) for 4 weeks, beginning at 4 weeks after surgery. $n = 5$ biologically independent samples per group in **c**. For MMP13 and ADAMTS5 staining, $n = 4$ mice (sham); $n = 5$ mice (DMM, DMM + agomir-17). For NOS2 staining, $n = 4$ mice per group. **e, f** qRT-PCR analysis of catabolic genes (**e**) and representative images of MMP13, ADAMTS5, and NOS2 immunostaining, and quantification of positive cells (**f**) from knee joints of cKO mice subjected to DMM surgery and intra-articular injection of agomir-NC, or agomir-17 (2 nmol). The injections were performed at 1 week after surgery and samples were collected at 2.5 weeks after surgery. $n = 4$ biologically independent samples per group in (**e**); $n = 4$ mice per group in (**f**). All scale bars, 100 μm. Data are presented as mean ± s.e.m. or boxplots (center line, median; box limits, 25 to 75th percentiles; whiskers, min to max), and dots represent individual mice or biologically independent samples. NS, not significant, *$P < 0.05$, **$P < 0.01$, and ***$P < 0.001$ in (**a, c–f**). *$P < 0.05$ versus control group, °$P < 0.05$ versus IL-1β + mimic NC in (**b**). Two-sided Student's $t$ test for (**a**); one-way ANOVA with Bonferroni's test for (**b–f**). Exact $P$ values in (**b**): *$P = 0.0011$, °$P = 0.0205$ (MMP3); *$P < 0.0001$, °$P = 0.0039$ (MMP13); *$P = 0.0015$, °$P = 0.0040$ (ADAMTS5); *$P = 0.0001$, °$P = 0.0013$ (NOS2); *$P = 0.0013$, $P > 0.9999$ (SOX9); *$P = 0.0001$, $P > 0.9999$ (COL2A1). Source data are provided as a Source Data file.

injections could induce higher and more sustained levels of miR-17 expression (Supplementary Fig. 6d), perhaps due to gradually improved cartilage micro-environment. In *miR-17~92* cKO DMM mice, GDF-5 administration failed to suppress the catabolic factors or prevent cartilage destruction (Supplementary Fig. 6e, f), suggesting that GDF-5 executed protective function against OA, largely via induction of miR-17 expression.

**_MiR-17~92_ deficiency in chondrocytes triggered cartilage degeneration without DMM.** In *miR-17~92* cKO mice, noticeable weakness of proteoglycan staining in the weight-bearing area of the femoral condyle was observed in knee joints at 4 weeks after tamoxifen injection even without DMM surgery. By 8 weeks, the affected area had expanded to cover nearly half of the joint surface, extended into the calcified layer (Fig. 4a, b). The pathology featured a marked decrease in cellularity of the superficial layer (70.1% fewer cells than in control mice at 4 weeks), matrix edema (increased cartilage thickness), and splitting (matrix separation)[26] in middle cartilage (Fig. 4a, b). Unexpectedly, neither staining of MMP13, ADAMTS5 and NOS2 nor the mRNA levels of the pathological catabolic factors showed significant upregulation in degenerated cartilage of cKO mice without DMM surgery (Fig. 4c, d), indicative of a different degenerative mechanism. Yet, intra-articular administration of agomir-17 for 4 weeks still demonstrated robust protective effect against cartilage degeneration in *miR-17~92* cKO mice (Fig. 4a, b), likely via a mechanism other than inhibition of the pathological catabolic factors.

**MiR-17 maintains cartilage homeostasis potentially via restriction of HIF-1α levels.** It has been well recognized that the joint cartilage is composed of four distinct layers and chondrocytes showed substantial heterogeneity in cellular morphology, arrangement, and matrix production[27]. However, little is known about the difference at the whole transcriptomic levels among the chondrocyte subsets. Additionally, changes in gene transcription of each chondrocyte subset under OA pathology remain to be fully elucidated. To explore the underlying mechanisms by which *miR-17~92* deficiency led to cartilage degeneration and/or how miR-17 reversed the phenotype, we performed scRNA-seq on cells isolated from the medial femoral condyle (Fig. 5a). After unsupervised hierarchical clustering, we identified a cluster with high expression of *Sox9*, *Acan*, *Comp* and *Fn* as a chondrocyte population (Supplementary Fig. 7a), which could be subsequently subdivided into three clusters (Chondrocyte1 (C1), C2 and C3) (Fig. 5b).

*Mmp2*, *Col14a1* and *Col22a1*, which are reported to be expressed mainly in the superficial zone[5,27,28], were highly enriched in clusters C1 (*Mmp2* and *Col14a1*) or C2 (*Col22a1*). In addition, C1 cluster also expressed *Serpinf1*, *Lum* and *Apod*; C2, *Clic5*, *Itga6*, and *Hbegf*; and C3, *Cytl1*, *Clu*, *Col2a1*, and *Cilp* (Fig. 5c). Immunofluorescent staining revealed distribution of MMP2, APOD, CLIC5, and COL22A1 on the cartilage surface and CYTL1 expressed only in the middle zone (Fig. 5d). We thus inferred that clusters C1 and C2 representing superficial chondrocytes, while cluster C3 representing middle zonal chondrocytes. *Prg4* (lubricin), a known superficial chondrocyte marker gene, was highly expressed in cluster C2 and was unexpectedly expressed also in cluster C3 (Supplementary Fig. 7a). It is likely that the *Prg4* mRNA is not translated into proteins or that the protein is degraded in middle zonal chondrocytes. Such a mode of step-wise shutting down of a gene has been reported in embryonic stem cells committed to neuronal fate[29]. By GO analysis, cluster C1 was enriched for GO terms featured in collagen fibril organization (*Col3a1*), and collagen catabolic process (*Mmp2* and *Mmp14*), suggesting high rates of collagen turnover. Cluster C2 was enriched with genes related to epithelial cell migration, which could be linked to lineages of the synovial lining layer[30]. Cluster C3 expressed genes associated with cartilage development (*Comp*) and collagen fibril organization (*Col2a1*, *Col11a1*) (Fig. 5e). In addition, biological processes of C1 and C3 were concentrated on cartilage-bone morphogenesis, but those of C2 focused on "neural crest cell migration" (Supplementary Fig. 7b). Thus, the superficial zone of cartilage may contain two previously unknown intermingled subpopulations of cells, both of which might be decreased in *miR-17~92* cKO mice.

We further analyzed scRNA-seq data from control, cKO, and agomir-17 rescued cKO mice. Consistent with histological analysis (Fig. 4b), the percentage of superficial zone C1 and C2 cells in the harvested osteochondral unit were dramatically reduced in cKO mice. Agomir-17 injection reversed abundances of C1 and C2 subsets to 105.3% and 60.4% of the control values, respectively. However, the abundance of cluster C3 remained relatively unchanged upon *miR-17~92* deletion (Fig. 6a). These data suggested that the numbers of C1 and C2 superficial zone cells were more sensitively fine-tuned by levels of miR-17. Gene expression analyses further demonstrated that in all three subpopulations of cells, miR-17 administration could pretty much revert altered gene expression due to *miR-17-92* cKO, back to control levels (Fig. 6b). Additional GO analysis of miR-17 dependent genes in all three cell populations included "collagen fibril organization", whereas "ECM organization" and "cartilage

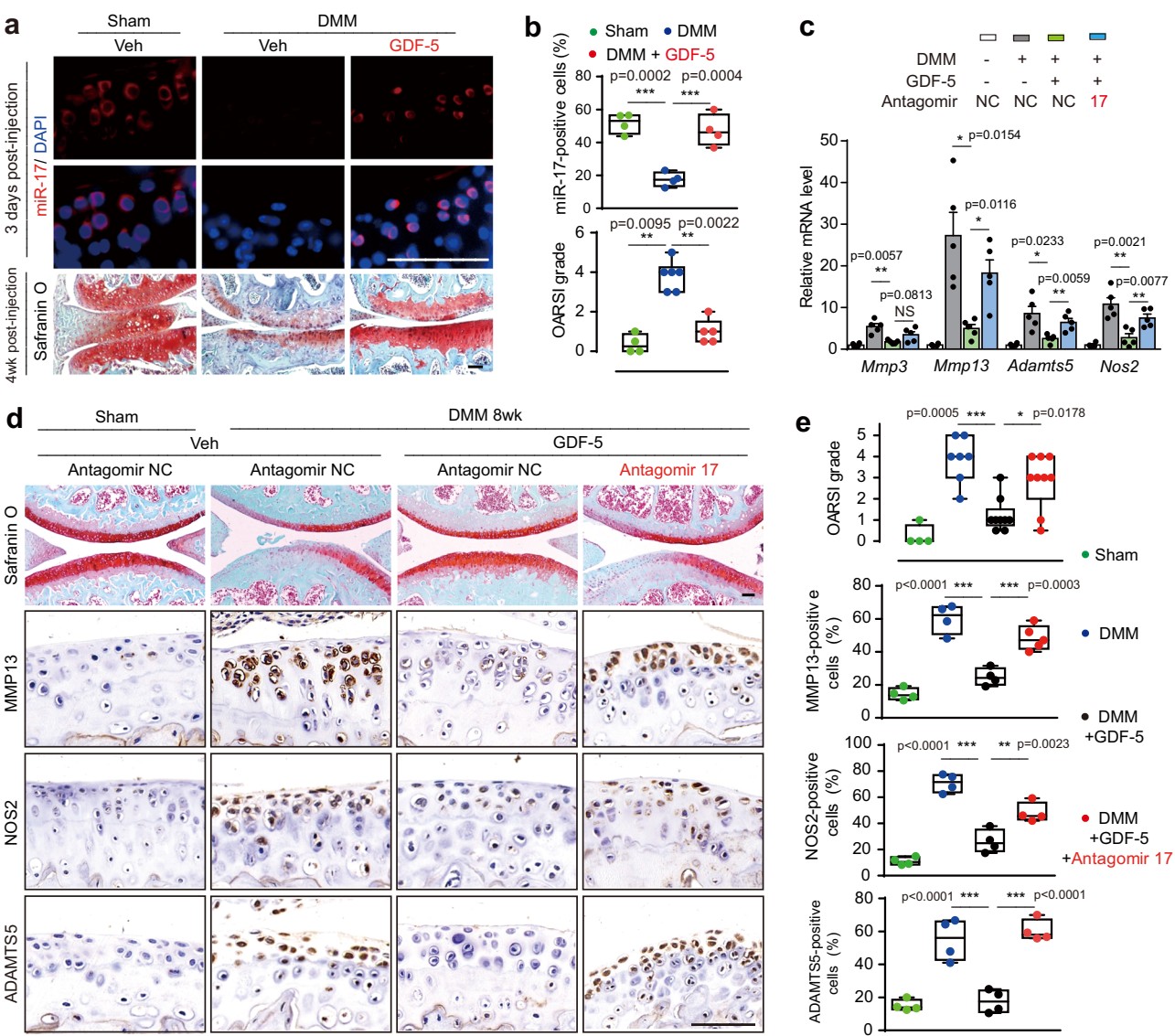

**Fig. 3 Endogenous miR-17 was upregulated by GDF-5 to suppress cartilage destruction in DMM mice. a, b** FISH for miR-17 and safranin O/fast green staining of joint sections (**a**), and quantification of miR-17-positive cells and OARSI scores (**b**). Veh (PBS) or GDF-5 (100 ng per injection) was injected weekly for 4 weeks, beginning at 4 weeks after sham or DMM surgery. $n = 4$ mice per group for cell counting. For OARSI grade, $n = 4$ mice (sham); $n = 6$ mice (DMM); $n = 5$ mice (DMM + GDF-5). **c–e** qRT-PCR analysis of catabolic genes (**c**); safranin O/fast green staining and IHC staining for MMP13, ADAMTS5 and NOS2 (**d**); OARSI scores; and quantification of cells positive for immunostaining (**e**) from knee joints. A combination of GDF-5 (0 or 100 ng) and 3 nmol of the antagomir (antagomir-17 or antagomir-NC) was injected intra-articularly. The injections were performed weekly for 4 weeks, beginning at 4 weeks after sham or DMM surgery. $n = 4$ biologically independent samples (sham + antagomir-NC) and $n = 5$ biologically independent samples for other groups in (**c**). For OARSI scores, $n = 4$ mice (sham); $n = 7$ mice (DMM); $n = 9$ mice (GDF-5 + antagomir-NC, GDF-5 + antagomir-17). For MMP13 staining, $n = 5$ mice (antagomir-17); $n = 4$ mice in other groups. For NOS2 and ADAMTS5 staining, $n = 4$ mice per group. All scale bars, 100 μm. Data were presented as boxplots or mean ± s.e.m. Boxplots: center line, median; box limits, 25th to 75th percentiles; whiskers, min to max. *$P < 0.05$, **$P < 0.01$, and ***$P < 0.001$. One-way ANOVA with Bonferroni's test for percentage of positive cells; two-sided Mann–Whitney U test for OARSI grade; two-sided Student's t test for (**c**). Source data are provided as a Source Data file.

development" were enriched in C1 and C3 (Fig. 6c). A comprehensive downregulation of collagen expression (*Col3a1* in C1; *Col1a1* and *Col22a1* in C2; *Col2a1* in C3) was observed in cKO mice (Supplementary Fig. 8), which was validated by immunohistochemical analyses (Supplementary Fig. 9). In accordance with this, human OA cartilage samples demonstrated loss of COL1A1 in cartilage surface and decreased COL2A1 expression in undamaged region, which may represent early OA pathology (Supplementary Fig. 10). Importantly, miR-17 expression was also significantly reduced in human OA cartilage (Fig. 6d), indicative of the human relevance of our findings.

We mapped the expression patterns of both anabolic and catabolic genes in all three clusters (Supplementary Fig. 8). Normal C1 chondrocytes expressed high levels of *Mmp2* and *Mmp14*; but *Mmp3*, *Mmp13* and *Adamts5* were not highly expressed in three clusters. With deletion of *miR-17~92*, without DMM, the expression of these catabolic genes did not increase, instead, *Mmp2* and *Mmp14* decreased expression. Importantly, expression of anabolic genes including *Fgfr1, Bmp2, Acan, Fn1, Col14a1, Prg4,* and *Loxl2* were also decreased in different cell populations, indicative of reduced anabolic process leading to cartilage degeneration. Intra-articular supplementation of agomir-17, restored the overall balance

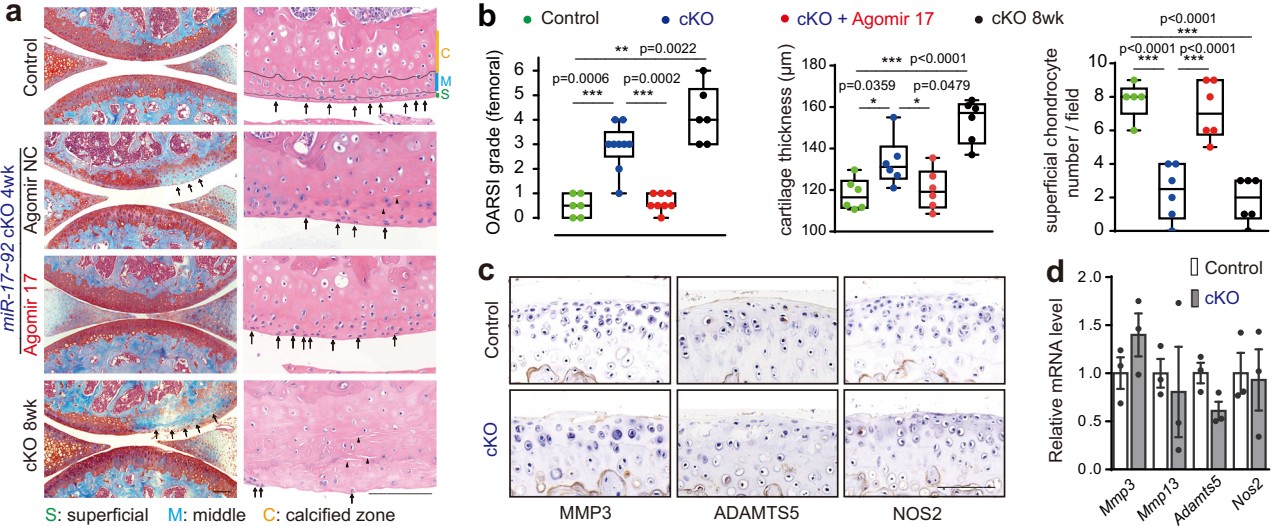

**Fig. 4 Cartilage degeneration was developed in *miR-17-92* cKO mice. a**, **b** Safranin O/fast green staining and H&E staining of knee joint cartilage (**a**), OARSI grade (*n* = 6 mice in control and cKO 8wk; n = 9 mice in cKO; n = 8 mice in cKO + agomir-17), cartilage thickness (*n* = 6 mice per group) and cell counts of superficial chondrocytes (*n* = 5 mice in control; *n* = 6 mice for other groups) in femoral condylar cartilage (**b**) from *miR-17-92^fl/fl* (control), *miR-17-92* cKO and agomir-17-injected cKO mice at 4 or 8 weeks after tamoxifen injection, were presented. The arrows in (**a**) indicated cartilage degeneration (left panels) and superficial chondrocytes (right panels). The triangles in (**a**) indicated splitting in cartilage matrix. **c**, **d** IHC staining of MMP13, ADAMTS5 and NOS2 (**c**), and qRT-PCR analysis of catabolic factors (**d**) in cartilage of control and *miR-17-92* cKO mice at 4 weeks after tamoxifen injection. *n* = 4 mice per group in (**c**) and *n* = 3 mice per group in (**d**). Data were presented as boxplots (center line, median; box limits, 25 to 75th percentiles; whiskers, min to max), or the mean ± s.e.m. values. *$P < 0.05$, and ***$P < 0.001$. Mann–Whitney *U* test (two sided) for OARSI grade; one-way ANOVA with Bonferroni's test for cartilage thickness and cell number; two-sided Student's *t* test for qRT-PCR. All scale bars, 100 μm. Exact *P* values in (**d**): 0.2236 (*Mmp3*), 0.7143 (*Mmp13*),0.0513 (*Adamts5*), 0.8648 (*Nos2*). Source data are provided as a Source Data file.

between anabolism and catabolism, and prevented cartilage degeneration. In scRNA-seq data, we found that genes for collagen-modifying enzymes were abundantly increased in cKO chondrocyte populations, including those of collagen prolyl hydroxylase (*P3h3*), lysine hydroxylase (*Plod2*) and lysyl oxidase (*Lox*), two of which were inhibited following agomir-17 injection (Fig. 6e). Simultaneously, expressions of genes for collagen were reduced (Supplementary Fig. 8). Reduced collagen synthesis and increased collagen modifications were also reported previously in growth plate cartilage of *Phd2^chon-* mice with accumulation of the hypoxia-inducible factor HIF-1α[31], allowing us to speculate that HIF-1α might play critical roles in dysregulation of collagen turnover in *miR-17~92* cKO mice. We thus examined expression levels of HIF-1α in joint cartilage of wild type and cKO mice and found that HIF-1α protein levels were elevated in cKO cartilage, along with decreased expression of SOX9, COL2A1, and MMP2. With administration of agomir-17, changes in the expression of HIF-1α, SOX9, COL2A1, and MMP2 were reversed (Fig. 6f). In cultured mouse chondrocytes treated with PHD2 inhibitor (IOX2 at 20 μM) for 72 h, HIF-1α was elevated and levels of *P3h3*, *Plod2* and *Lox* were increased. While the expression of SOX9 and MMP2 were decreased. Treatment with combination of miR-17 inhibitor and IOX2 caused increased expression of HIF-1α and collagen-modification enzymes, and further decreases in SOX9 and MMP2 expression. In contrast, overexpression of miR-17 resulted in significant decreases of HIF-1α, along with increased expression of SOX9, MMP2 and decreased expression of collagen-modification enzymes (Fig. 6g–i). However, we did not observe significant changes in COL2A1 protein expression upon IOX2 treatment, probably because increased collagen-modification protected COL2A1 from degradation. This is slightly different from the in vivo situation, potentially due to lack of mechanical loading in culture conditions. Given that the target site of miR-17 in 3′UTR of HIF-1α has been well characterized in previous report using miRNA-reporter assays[32], we conclude that miR-17 maintains

cartilage homeostasis potentially through directly targeting HIF-1α, by which collagen turnover was sustained in balance under physiological circumstances.

## Discussion

In this study we showed that miR-17, which is highly expressed in superficial and middle zones of the articular cartilage, protected against DMM-induced cartilage destruction by simultaneously targeting pathological catabolic factors, including MMP3, MMP13, ADAMTS5, and NOS2. We also discovered that GDF-5 upregulated endogenous miR-17 expression to exert its protective function against OA. ScRNA-seq analyses of *miR-17~92* knock-out chondrocytes and further experiments revealed that, miR-17 also positively regulated physiological catabolic and anabolic factors involved in cartilage homeostasis potentially via restriction of HIF-1α accumulation (Fig. 7).

The *miR-17~92* cluster contains 6 different microRNA members. Together, they target more than 30 genes, which are known to be involved in diseases of the nervous, cardiovascular, and skeletal systems[33]. In the current study, we have demonstrated critical roles of miR-17 in regulating cartilage homeostasis, although we do not exclude the possibility that miR-92, miR-20a, or even miR-19 also participate in the regulation to certain extent. In DMM-induced OA, major catabolic factors including MMP3/13, ADAMTS5, NOS2 are upregulated and function to destroy cartilage in the pathological condition. During OA, endogenous miR-17 is downregulated, leading to "loss of protection" of the cartilage. MiR-17, through simultaneously and directly target 3′ UTRs of these genes, effectively suppress the function of these pathological catabolic factors. However, the severity of cartilage destruction in DMM mice with antagomir-17 injections was less than that in DMM cKO mice. Knockdown of endogenous miR-17 by injecting antagomir-17, unlike deletion of the whole *miR-17~92* cluster, failed to trigger a spontaneous cartilage

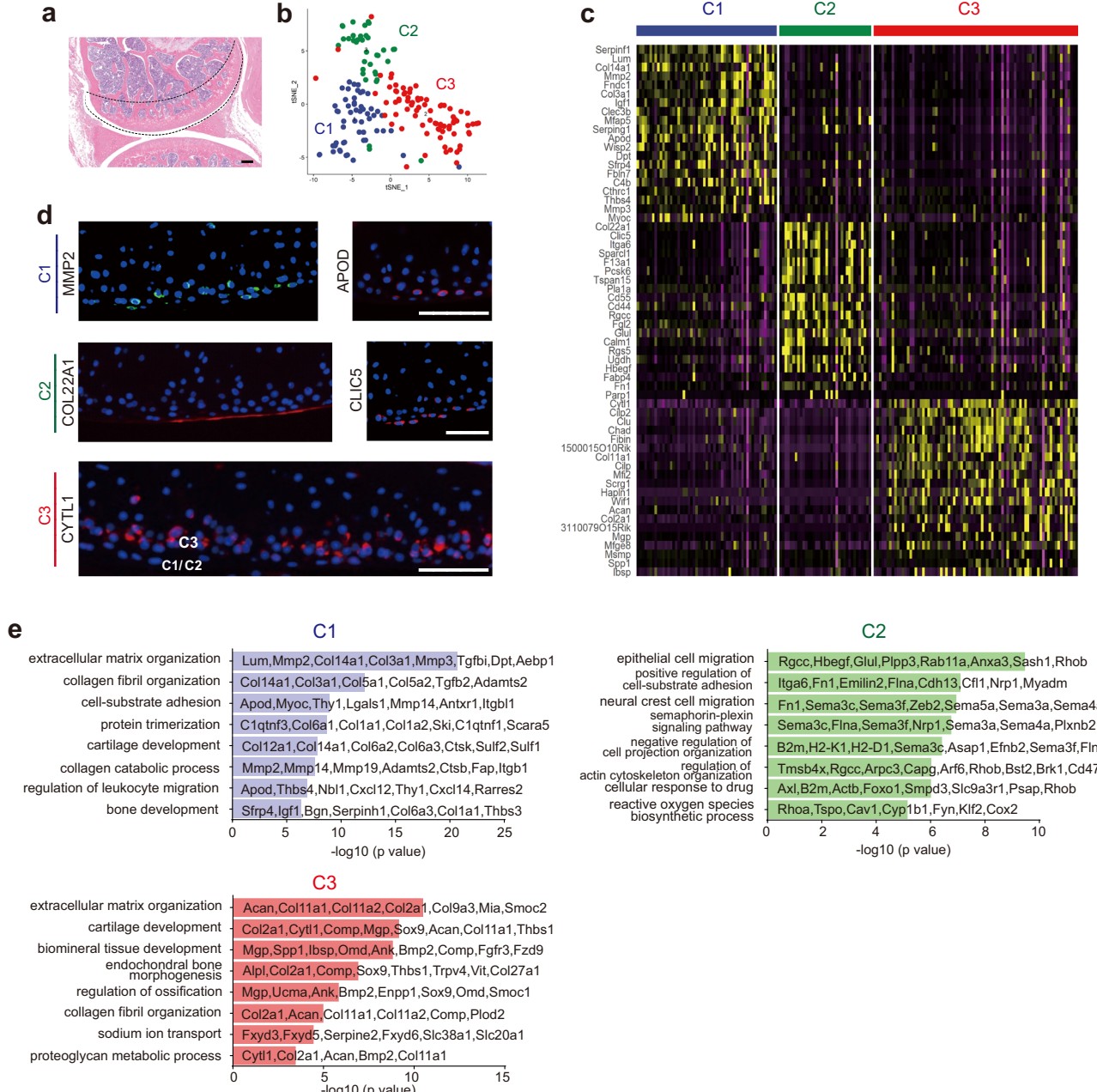

**Fig. 5 scRNA-seq analysis identified three subsets of chondrocytes. a** Medial femoral condyle tissue from the region enclosed by the dotted lines was isolated for cell dissociation. $n = 9$ mice. Scale bar, 250 μm. **b** Putative chondrocyte population (171 cells) and unsupervised clustering identified clusters C1, C2, and C3 in cartilage. **c** Heatmap showing the expression of the top 20 identified significantly differentially expressed genes in clusters C1, C2, and C3. **d** Immunofluorescence staining of representative markers identified by scRNA-seq in articular cartilage. $n = 3$ independent experiments. All scale bars, 100 μm. **e** GO analysis and representative genes for clusters C1, C2, and C3.

degeneration. These results indicated functional cooperation among members of *miR-17~92* cluster and they might share similar targets. For example, miR-92 has been reported to directly target ADAMTS5 in chondrocytes[34], and miR-19 was also predicted to target HIF-1α.

The superficial layer of the articular cartilage is the first to sense physio-pathological stimuli and response to mechanical loadings. ScRNA-seq and subsequent immunohistochemical analyses demonstrated that the superficial zone of the cartilage contained two intermingled cell subpopulations with one carrying features of synovial lining layer expressing type XXII collagen and lubricin (*Prg4*), and the other expressing types I, III, IV, VI, XIV collagen, etc. These collagens function to reinforce cartilage structural

network by crosslinking collagen fibrils, mediating intermolecular interactions and coordinating cell-matrix interactions[35]. Recent report using scRNA-seq of human osteoarthritic cartilage also indicated the presence of chondrocyte subtypes[36]. Together these observations demonstrated that cellular heterogeneity existed not only across different zones but also within a single layer of the cartilage. The superficial zone was found to be a structurally important element, which greatly limits swelling of the entire cartilage layer[37,38] by maintaining fluid pressurization within the tissue[39]. Thus, the decrease of superficial chondrocytes in cKO cartilage and subsequent degeneration of superficial zone contributed to matrix edema, leading to increases of cartilage thickness. Superficial chondrocytes appeared to be more sensitive to

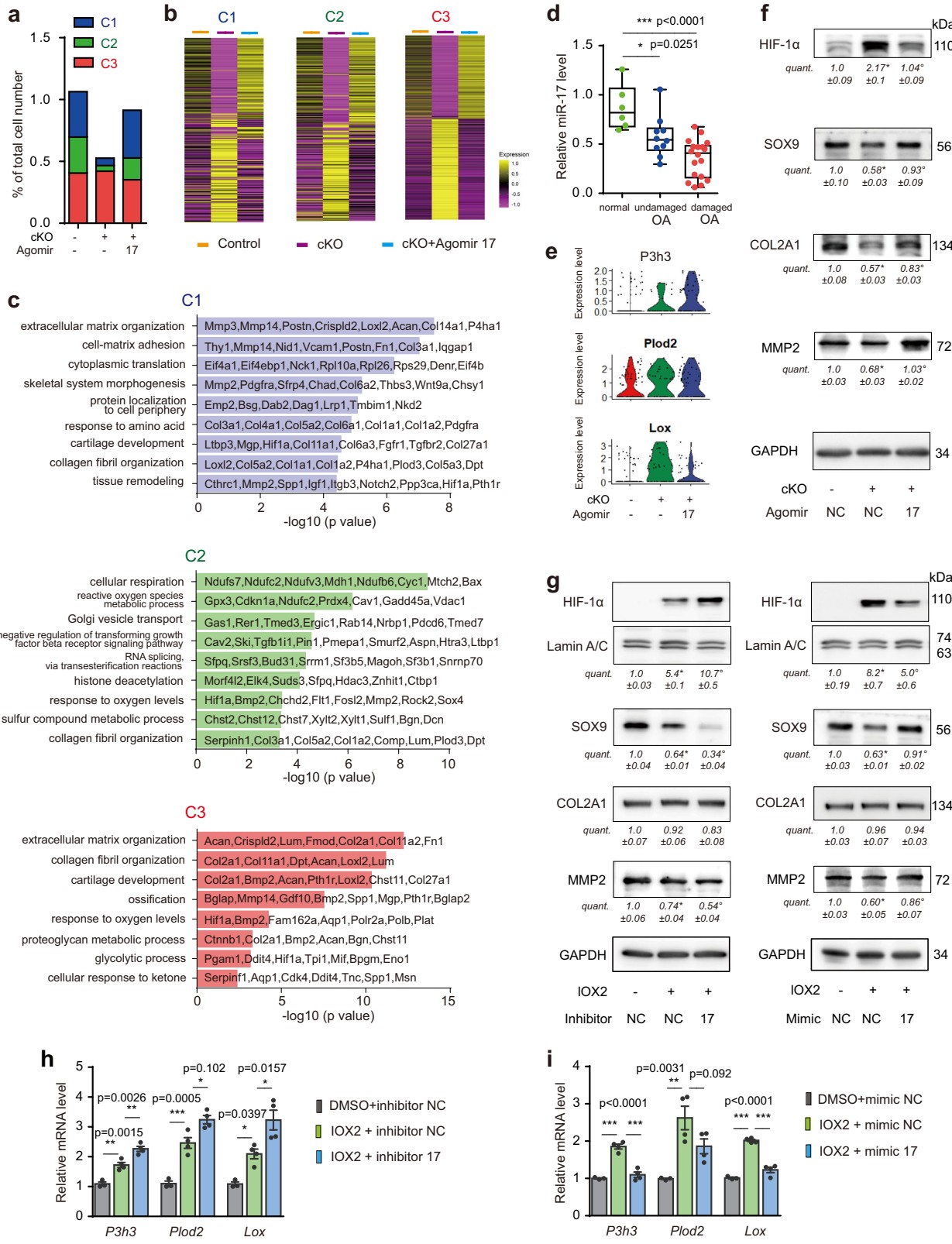

*miR-17~92* deletion than middle layer chondrocytes, demonstrating the importance of miR-17 expression in superficial chondrocyte homeostasis. Moreover, following DMM surgery, the dramatic drop of miR-17 positive cells in the superficial layer provided an important indicator and a cause for subsequent OA development.

To maintain cartilage homeostasis, a continuously ongoing degradation of old matrix is critical for deposition of newly synthesized matrix molecules[7]. In this study, we found MMP2, 14, and ADAMTS2 were expressed by healthy superficial chondrocytes as physiological catabolic factors, while MMP3, 13, and ADAMTS5 were not involved. In fact, it has been reported that a

**Fig. 6 MiR-17 maintains balanced anabolism and catabolism in subsets of articular chondrocytes. a** Cell percentages in each cluster of control, *miR-17~92* cKO and cKO + agomir-17 mice. **b** Genes reversed by miR-17 supplementation in *miR-17~92* cKO mice in clusters C1, C2, and C3. **c** GO analysis of genes upregulated in clusters C1, C2, and C3 in *miR-17~92* cKO+ agomir-17 mice in (**b**). **d** qRT-PCR analysis of miR-17 expression in normal ($n = 6$), undamaged ($n = 10$) osteoarthritic and damaged ($n = 18$) osteoarthritic human cartilage. **e** Violin plots showing changes in levels of collagen-modifying genes in control, *miR-17~92* cKO and cKO + agomir-17 mice. **f** Protein levels of HIF-1α, SOX9, COL2A1, and MMP2 in articular cartilage of control, *miR-17~92* cKO and cKO + agomir-17 mice at 2 weeks after tamoxifen injections were determined by western blot followed by densitometry analysis. Blots are representative of three independent experiments. **g–i** Mouse articular chondrocytes were transfected with miR-17 inhibitor (100 nM) or inhibitor NC or miR-17 mimic (50 nM) or mimic NC and treated with or without IOX2 (20 µM) for 72 h. The protein levels of HIF-1α, SOX9, COL2A1 and MMP2 were determined by western blot followed by densitometry analysis. Blots are representative of three independent experiments (**g**). The gene expression of *P3h3*, *Plod2*, and *Lox* were determined by qRT-PCR. $n = 3$ biologically independent samples (none-IOX2 treatment groups); $n = 4$ biologically independent samples (IOX2 treatment groups) (**h**, **i**). Data are presented as means ± s.e.m. or boxplots (center line, median; box limits, 25 to 75th percentiles; whiskers, min to max), and dots represent biologically independent samples. *$P < 0.05$, **$P < 0.01$, and ***$P < 0.001$ in (**d**, **h**, **i**); *$P < 0.05$ versus control group, °$P < 0.05$ versus cKO group in (**f**); *$P < 0.05$ versus none-IOX2 group, °$P < 0.05$ versus IOX2 with inhibitor/mimic NC group in (**g**) (one-way ANOVA with Bonferroni's test). Exact $P$ values in (**f**): *$P = 0.0004$, °$P = 0.0005$ (HIF-1α); *$P = 0.0219$, °$P = 0.0467$ (SOX9); *$P = 0.0026$, °$P = 0.0303$ (COL2A1); *$P = 0.0003$, °$P = 0.0002$ (MMP2). Exact $P$ values in left panel of (**g**): *$P < 0.0001$, °$P < 0.0001$ (HIF-1α); *$P = 0.0009$, °$P = 0.0025$ (SOX9); $P = 0.9211$, $P = 0.7283$ (COL2A1); *$P = 0.0162$, °$P = 0.0454$ (MMP2). Exact $P$ values in right panel of (**g**): *$P = 0.0002$, °$P = 0.0140$ (HIF-1α); *$P < 0.0001$, °$P = 0.0001$ (SOX9); $P > 0.9999$, $P > 0.9999$ (COL2A1); *$P = 0.0024$, °$P = 0.0201$ (MMP2). Source data are provided as a Source Data file.

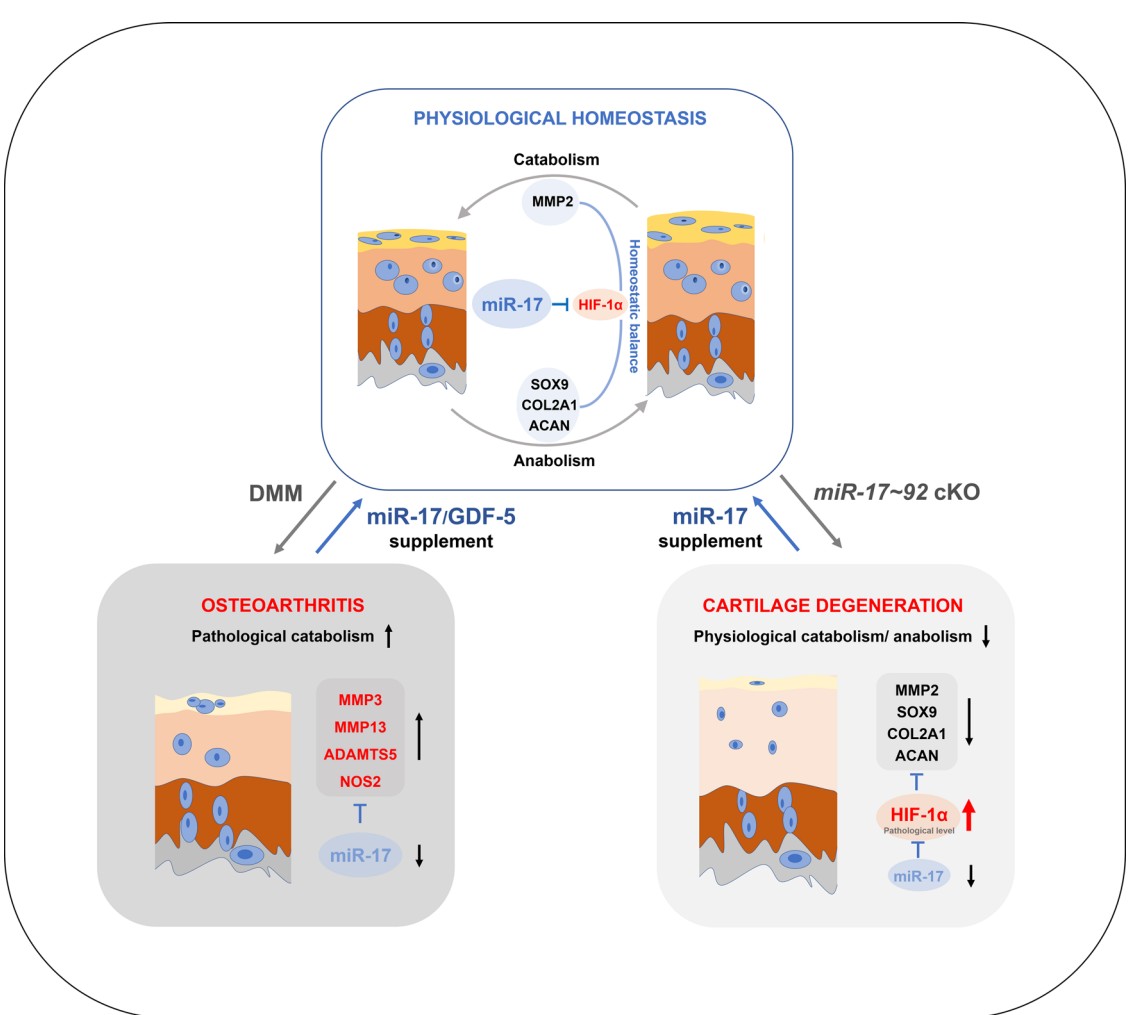

**Fig. 7 Model depicting the mechanisms of miR-17 in maintaining cartilage homeostasis and prevention of OA.** MiR-17 is a regulator for HIF-1α in articular chondrocytes, and catabolic and anabolic balance was maintained under physiological HIF-1α expression. The cellular and ECM change of the articular cartilage was shown in mouse models of DMM or *miR-17~92* cKO. The faded color in superficial and middle layers indicates ECM destruction or degeneration. In DMM-induced OA, pathological catabolic factors targeted by miR-17 were increased. In *miR-17~92* cKO mice, miR-17 deficiency caused pathological increase of HIF-1α, and decreased the physiological catabolism and anabolism in chondrocytes. Supplementation with exogenous miR-17 or inducing endogenous miR-17 by GDF-5 treatment prevented OA and restored cartilage homeostasis.

significant amount of MMP2 is expressed in the superficial zone of human normal cartilage[40]. In order to match with their highly anabolic metabolism, superficial chondrocyte undergoes robust catabolism to maintain active matrix turnover[41]. Interestingly, in *miR-17-92* cluster cKO mice, cartilage underwent degeneration without upregulation of the aforementioned pathological catabolic factors (MMP3,13, etc.), as revealed by scRNA-seq. Instead, miR-17 deficiency led to reduced expression of the physiological catabolic factors (*Mmp2,14*) as well as anabolic factors (*Fgfr1, Acan, Fn1, Col14a1,* and *Loxl2*), and supplementation of miR-17 reversed these changes. The alteration of cartilage matrix would significantly decrease the load bearing capacity of articular cartilage, further damaging the collagen and proteoglycan structure[42].

Despite the importance of HIF-1α in ECM synthesis[43], epiphyseal chondrocytes require PHD2-regulated inactivation of HIF-1α to avoid skeletal dysplasia, since pathological increase of HIF-1α signaling interferes with cellular bioenergetics and biosynthesis[31]. It is unknown whether there is a requirement for restriction of HIF-1α signaling in articular chondrocytes. Our data showed the HIF-1α level was enhanced, yet, anabolic as well as physiological catabolic factors were reduced in cKO chondrocytes as well as by prolonged treatment of PHD2 inhibitor. Overexpression of miR-17 caused decreased levels of HIF-1α and restored physiological catabolic and anabolic balance, showing miR-17 functions as a factor to control HIF-1α levels. The appropriate HIF-1α signaling is vital for chondrocytes metabolism and ECM production. It has been reported that any anomalous subtle alterations in anabolic reactions, as indicated in knockdown of Kdm6b[44] or interruption of TGF-β1 signaling[45], can result in initiation of homeostasis disruption of articular cartilages. Therefore, miR-17 could function under both physiological and pathological conditions, playing different roles in modulating the anabolic and catabolic activities in maintaining joint cartilage homeostasis as well as repair in OA.

In conclusion, our study revealed dual roles of miR-17 in maintaining cartilage homeostasis and prevention of OA, suggesting that miR-17 could be a potentially drug target for OA. Given that the therapeutic benefits of miRNAs in disease intervention have been increasingly documented in preclinical studies[46], supplementation of miR-17 might be a potent therapeutic approach for OA as well as prevention of cartilage aging.

## Methods

**Human specimens.** Normal human articular cartilage without arthritis was collected from the knee or hip joints from 6 patients with osteosarcoma or trauma who were undergoing surgery. Arthritic cartilage specimens were obtained from individuals with OA within 2 h after arthroplasty. OA was diagnosed according to the criteria of the American College of Rheumatology. The specimens were processed for qRT-PCR and histological examination. The patients without arthritis (1 male and 5 female) had an average age of 55.5 years. The patients with OA (4 male and 14 female) had an average age of 66.06 years. The collection of discarded human tissues was approved by the Research Ethics Board of Affiliated Hospital of Xuzhou Medical University. Written informed consent was provided by all patients involved in this study. The study design and conduct complied with all relevant regulations regarding the use of human study participants and was conducted in accordance with the criteria set by the Declaration of Helsinki.

**Mice.** Male C57BL/6J (wild type) mice were obtained from the Shanghai Animal Center and randomized into experimental groups. Mice were housed under ambient temperature of 24 ± 2 °C, circulating air, constant humidity of 50 ± 10% and a 12 h:12 h light/dark cycle. The animal sample size for each experiment was selected based on previous, well-characterized studies[47]. The number of animals in each group is specified in figure legends.

Floxed *miR-17~92fl/fl* mice[13] (stock no. 008458) and *Col2CreERT* mice[48] (stock no. 006774) were obtained from the Jackson Laboratory. To generate *Col2CreERT;miR-17~92fl/fl* mice (*miR-17~92* cKO), *miR-17~92fl/fl* mice were mated with *Col2CreERT* mice, and the resulting *Col2CreERT;miR-17~92fl/+* mice were then mated with *miR-17~92fl/fl* mice. We determined the genotype of the transgenic mice by PCR analysis of genomic DNA isolated from mouse tail snips. Genotyping for the loxP *miR-17~92*

allele was performed with the following primers: forward, 5′-TCGAGTATCTGACAATGTGG-3′ and reverse, 5′-TAGCCAGAAGTTCCAAATTGG-3′. The Cre transgene was identified with the primers 5′-CACTGCGGGCTCTACTTCAT-3′ and 5′-ACCAGCAGCACTTTTGGAAG-3′. The internal positive control primers were 5′-CAAATGTTGCTTGTCTGGTG-3′ and 5′-GTCAGTCGAGTGCACAG TTT-3′.

All animals were maintained in the Animal Facility of Tongji University School of Medicine. The experimental procedures involving mice were performed in accordance with the Guidelines for the Care and Use of Laboratory Animals and approved by the Animal Care and Experiment Committee of Tongji University.

**The surgeries and intra-articular injections for mice.** We anesthetized 10-week-old male mice with ketamine and xylazine. To induce mechanical instability-associated OA, DMM was surgically performed by transection of the medial meniscotibial ligament in the right knee joints, as previously described[49]. Control mice received a sham operation on the knee joint.

For the time-course study, operated knee joints were collected at 0, 4, 8, or 12 weeks after surgery. For antagomir (RiboBio Tech) delivery, mice were randomized into four groups. Mice were subjected to sham operation or DMM surgery and weekly intra-articular injection of antagomir-NC or antagomir-17 (3 nmol) by inserting a small needle into the sub-patellar region for three times from 1 week post-operation. For agomir delivery, mice were randomized into four groups. A total of 1.5 nmol of agomir-17 or agomir-NC (RiboBio Tech) in 8 μL of PBS was injected into the knee joints at 4 weeks after surgery. For the GDF-5 study, mice were randomized into 3 groups: 1) Sham, 2 and 3) at 4 weeks after DMM surgery, injected with 8 μL of recombinant mouse GDF-5 (100 ng; PeproTech) or the equivalent volume of vehicle (PBS) into knee joints. For antagomir delivery, mice were randomized into four groups. A combination of GDF-5 (0 or 100 ng) and 3 nmol of the antagomir (antagomir-17 or antagomir-NC, RiboBio Tech) in 8 μL of PBS was injected intra-articularly. Except where stated, agomir-, GDF-5-, and antagomir-treated mice received weekly injections for 4 weeks beginning at 4 weeks after surgery and were euthanized at 8 weeks after surgery. Knee joints were processed for histology and immunohistochemical (IHC) staining. Articular cartilage was harvested for mRNA analysis by qRT-PCR. Samples obtained from animals were excluded from further analysis if there was evidence of joint infection or if the animal died after the procedure. These exclusion criteria were established prior to the surgeries.

**The cKO mice induction and operations.** To activate the estrogen receptor (ER), tamoxifen (Sigma, 100 μg per gram of body weight) was intraperitoneally injected into 7-week-old *miR-17~92fl/fl* and *miR-17~92* cKO mice daily for 6 days[44]. To confirm knockout of miR-17, articular cartilage was isolated for qRT-PCR analysis at 1 week after tamoxifen injections. Then, sham or DMM operations were performed on 8-week-old *miR-17~92* cKO mice and their *miR-17~92fl/fl* littermates as described above to establish an experimental model of OA. The transgenic mice were euthanized at 4 or 8 weeks after tamoxifen injections, and knee joints were processed for histological and IHC staining. The intervention of agomir-17, agomir-19 and agomir-92 or GDF-5 in cKO mice were performed at 1 week post the initial tamoxifen injection, and analyzed at 4 weeks post-surgery.

**Quantitative RT-PCR (qRT-PCR).** Total RNA was extracted from cultured chondrocytes or mouse cartilage tissue with TRIzol Reagent (Life Technologies) according to manufacturer's instructions. Each in vivo cartilage sample represented pooled tissues from 2-3 knee joints. Human cartilage was homogenized with a TissueLyser-48L (Shanghai Jingxin), and RNA was isolated and purified using an RNeasy Fibrous Tissue Mini Kit (Qiagen). Then, 0.5 μg of RNA was reverse transcribed to cDNA using a PrimeScript RT Reagent Kit (Takara), and qPCR was performed in an ABI 7500 Fast System (Applied Biosystems). GAPDH or U6 was used for normalization. The reverse transcription and PCR primers specific for U6 and the miRNAs were purchased from RiboBio, and sequences of other PCR primers used in the study were listed in Supplementary Table 1.

**Histological, IHC, and FISH analyses.** Mouse knee joints were fixed with 4% paraformaldehyde for 48 h, decalcified with 10% EDTA for 10 days, and dehydrated for paraffin embedding. Serial sagittal or coronal sections were harvested at ~100 μm intervals to obtain 15–18 slides. The sections were cut to a thickness of 6 μm, dewaxed and stained with safranin O and fast green to visualize cartilage matrix proteoglycans or with hematoxylin and eosin (H&E) to visualize chondrocytes and matrix structures. To grade OA severity, OARSI scoring system[50] was performed by two observers blinded to the experimental groups to evaluate cartilage destruction in the medial joints. We scored the cartilage destruction from the medial tibial plateau in all DMM-involved models and medial femoral condyles in cKO mice without DMM surgery. For each joint, 3–5 sections were used for safranin O staining and the median score was representative for each biological samples. Human cartilage was cut off, fixed and embedded in paraffin and sectioned at 10-μm thickness. For IHC analysis, sections were incubated overnight at 4 °C with primary antibodies specific for MMP13 (1:400, 18165-1-AP; Proteintech), ADAMTS5 (1:200, DF13268; Affinity), NOS2 (1:200, PA3-030A; Pierce), COL1A1 (1:200, GB11022-1; Servicebio), COL2A1 (1:200, GB11021; Servicebio), COL3A1

(1:400, ab6310; Abcam), and COL22A1 (1:200, ab121846; Abcam). Staining was visualized using an HRP detection system (Servicebio), and sections were then counterstained with hematoxylin.

FISH for miR-17-5p was performed according to the standard protocol for sections[51], with minor modifications. In brief, joints were fixed, decalcified and submerged in sucrose solution, frozen in Optimal Cutting Temperature (OCT) compound and then sliced at a thickness of 10 μm. Sections were permeabilized with proteinase K for 5 min at 37 °C and treated with 3% $H_2O_2$ to block endogenous peroxidase activity. Slides were then incubated with the relevant probes at the concentration of 1 nM overnight in an airtight incubation chamber. Then, the signals were visualized using anti-digoxigenin-peroxidase (anti-DIG-POD) and incubated with Cy3-tyramide in 0.001% $H_2O_2$; DAPI staining was used to visualize cell nuclei. A DIG-labeled miR-17 oligo probe (mmu-miR-17-5p; YD00614506-BCG) was purchased from Qiagen. For immunofluorescence analysis, joints were frozen in OCT compound and sliced at a thickness of 8 μm. Sections were incubated with antibodies specific for MMP2 (1:500, GB11130; Servicebio), CLIC5 (1:200, ab66630; Abcam), APOD (1:300, DF7987; Affinity), COL22A1 (1:200, ab121846; Abcam) and CYTL1 (1:400, 15856-1-AP; Proteintech) overnight at 4 °C and were then incubated with Alexa Fluor 488- conjugated secondary antibody (1:300, A-11034; Invitrogen) or Cy3-conjugated secondary antibody (1:400, A10520; Invitrogen) for 30 min. For the histological and IHC analyses, images were scanned with a Pannoramic MIDI (3D HISTECH) and analyzed using CaseViewer 2.3 (3D HISTECH). FISH images were captured with a laser-scanning confocal microscope (True Confocal Scanner SP5, Leica) using the LAS AF Lite 2.6.0 software (Leica). For each joint, the percentage of positive cells was counted on 3–5 fields and the median percentage was representative for each mice.

**Culture of chondrocytes**. Cartilage was harvested from the knee joints of 7-day-old C57BL/6J mice and digested with 0.2% NB4 collagenase (Serva). Isolated chondrocytes (passage 0–1) were cultured at 37 °C in Dulbecco's modified Eagle's medium (Invitrogen) supplemented with 10% fetal bovine serum (FBS), 100 units/mL penicillin and 100 μg/mL streptomycin (Gibco). All treatment with mouse recombinant IL-1β (PeproTech) was at 5 ng/mL, IOX2 (MedChemExpress) was at 20 μM or DMSO as control.

**MiRNA overexpression or knockdown**. The miR-17, miR-19 and miR-92 microRNA mimic (50 nM final concentration) and miR-17 inhibitor (100 nM final concentration) constructs were obtained from RiboBio and transiently transfected into chondrocytes using Lipofectamine 2000 (Life Technologies) according to manufacturer's recommendations. A nonsense scrambled miRNA sequence was used as the negative control. After 6 h, the medium was replaced with serum-free medium, and cells were used for further studies 6 h later.

**Western blot**. For NOS2, SOX9, and COL2A1 protein detection, chondrocytes were lysed in RIPA buffer containing a protease inhibitor cocktail (Merck). For secreted protein detection, the cell culture medium was concentrated in ultracentrifuge tubes (Millipore) at 7000 rpm for 30 min. For nuclear HIF-1α detection, nuclear extracts were prepared using a Nuclear Protein Extraction Kit (Beyotime). For protein detection from murine articular cartilage, tissues were dissected using microsurgical scissors, and a polled sample was from four knee joints. Protein concentrations were determined with BCA Protein Assay Reagent (Thermo Fisher Scientific). Extracted proteins (20 μg) were separated by SDS-PAGE and electro-transferred to polyvinylidene fluoride membranes (Bio-Rad Laboratories). After blocking with 5% bovine serum albumin (Sigma-Aldrich), membranes were incubated with the following primary antibodies at 4 °C overnight: anti-MMP3 (1:1000, rabbit monoclonal [EP1186Y], ab52915; Abcam), anti-MMP2 (1:1000, rabbit polyclonal, GB11130; Servicebio), anti-MMP13 (1:2000, rabbit polyclonal, 18165-1-AP; Proteintech), anti-ADAMTS5 (1:200, rabbit polyclonal, ab41037; Abcam), anti-NOS2 (1:1000, rabbit monoclonal [EPR16635], ab178945; Abcam), anti-SOX9 (1:1000, rabbit monoclonal [EPR14335-78], ab185966; Abcam), anti-COL2A1 (1:1000, rabbit polyclonal, 28459-1-AP; Proteintech), anti-HIF-1α (1:500, rabbit monoclonal [EPR16897], ab179483; Abcam), Lamin A/C (1:2000, mouse monoclonal [4C11], 4777S; Cell Signaling Technology), and GAPDH (1:1000, mouse monoclonal [2F40], T0004; Affinity). After washing, the membranes were incubated with horseradish peroxidase (HRP)-conjugated secondary antibodies (goat anti-mouse IgG, 1:5000, 115-035-003, and goat anti-rabbit IgG, 1:5000, 111-035-003; Jackson Immunoresearch) for 30 min at 37 °C, and signals were detected by enhanced chemiluminescence.

**MiRNA target analysis and luciferase assays**. To predict miR-17 targets, miRanda and TargetScan were used to scan the 3′-UTRs of murine mRNAs. For the 3′-UTR reporter assay, the *Mmp3*, *Mmp13*, *Adamts5*, *Nos2* and *Epas1* 3′-UTRs containing the miR-17 target sequences were chemically synthesized by Sangon Biotech and cloned into the psiCHECK-2 vector (Promega). Site-specific mutations were introduced in the miR-17 binding sequence, and the sequence was cloned by PCR into the psiCHECK-2 vector. HEK293T cells (ATCC, CRL-11268; tested by PCR for mycoplasma contamination) were cotransfected with 10 pmol of the miRNA mimic and 10 ng of the 3′-UTR luciferase reporter vector per well in 96-well plates in triplicate using Lipofectamine 2000 (Invitrogen). Cells were lysed 48 h after transfection, and luciferase activity was determined by a Dual-Luciferase Reporter Assay (Promega). Firefly luciferase activity was normalized to *Renilla* luciferase activity. The sequences of the PCR primers used to introduce site-specific mutations are listed in Supplementary Table 2.

**Droplet-based scRNA-seq**. A total of nine littermate mice, including *miR-17~92* cKO, *miR-17~92fl/fl* and agomir-17-treated *miR-17~92* cKO mice, were used in this experiment. Mice in the agomir-17 group received weekly injections of agomir-17 (2 nmol/injection) in the knee joints for a total of 4 injections. After mice were euthanized, the medial femoral condyle, including the articular cartilage, sub-chondral bone and adjacent bone marrow cavity, was dissected using scissors to ensure that intact cartilage tissue was obtained. The tissue was cut into small pieces and was then digested at 37 °C for 2 h in 10 mL of DMEM containing 0.2% NB4 collagenase and 0.3 mg/mL DNase I (Thermo Scientific). The mixture was filtered through a 40-μm strainer and centrifuged at 300 × g for 10 min. Cells were counted using a hemocytometer, and ~8000 cells in 25 μL of PBS per group were used for experiments. Single cells were processed on a Chromium Single-Cell Platform with a Chromium Single-Cell 3′ Library and Gel Bead Kit v2 (10X Genomics, PN-120237) and a Chromium Single-Cell A Chip Kit (10X Genomics, PN-120236) according to the manufacturer's protocol. DNA quality was assessed with an Agilent Bioanalyzer to ensure transcript integrity and purity, and amount was assessed with a Qubit fluorometric assay (Invitrogen).

**Preprocessing of scRNA-seq data**. Raw sequencing data were aligned to the mm10 (Ensembl 84) reference genome, and estimated cells and associated unique molecular identifiers (UMIs) were counted using the CellRanger Single-Cell Software Suite (version 2.1.1, 10X Genomics). Downstream analyses were performed with the R software package Seurat version 2.3.4[52]. First, genes expressed in <4 cells and cells expressing <200 genes were excluded. Then, the UMIs were log normalized for each cell using the natural logarithm of 1 + UMIs per 10,000. Variable genes were selected with the *FindVariableGenes* function using thresholds for average expression (>0.005 and <3) and dispersion (0.3). Variation in UMI counts between cells was regressed using the *ScaleData* function with default parameters. Principal component analysis was performed on the regressed highly variable genes with "RunPCA" (pcs.compute = 100), and the top 100 principal components were used to conduct fast Fourier transform-accelerated interpolation-based *t*-distributed stochastic neighbor embedding (FIt-SNE). The cells were then clustered using the function "FindClusters" (resolution = 1.2, n.start = 10, nn.eps = 0.5, k.param = 20). For each cluster, differentially expressed genes were identified using the "FindAllMarkers" or "FindMarkers" functions with default parameters. The clusters that expressed several chondrocyte markers at high levels were selected for downstream analysis. Single-cell gene expression data were analyzed for differential gene expression across the chondrocyte subsets. Differential gene expression was determined using UMI-normalized gene expression data. The Reactome, KEGG and GO Biological Functions libraries were used to analyze the scRNA-seq data. Enrichment of GO terms in the "biological process" category was conducted for differentially expressed genes (log (fold change) ≥0.5 or <−0.5) using the Bioconductor GOstats package. The top GO terms were filtered by size (between 30 and 300 genes) to select for terms that were neither excessively general nor specific.

**Statistical analysis**. The assessment of OARSI grades and count of immunostaining positive cells were performed in a blinded manner. Statistical significance was assessed using an unpaired, two-tailed Student's *t* test for comparisons between two groups to evaluate qRT-PCR data, luciferase assay data and the percentages of positive cells in tissue sections. The variance was estimated within each group of data. The equivalence of variance was confirmed using an F test for comparisons between two groups. For multigroup comparisons, one-way ANOVA followed by a post hoc test was used. For nonparametric data, the Mann–Whitney *U* test was used. For nonparametric, multigroup comparisons, the Kruskal–Wallis test followed by Dunn's multiple comparisons test was used. All analyses were performed with GraphPad Prism 6 software (GraphPad Software). P < 0.05 was considered statistically significant.

**Reporting summary**. Further information on research design is available in the Nature Research Reporting Summary linked to this article.

## Data availability

Single-cell RNA-seq files have been uploaded to the websites, Accession: SRX10016273, SRX10016274, SRX10016275. Raw sequencing data were aligned to the mm10 (Ensembl 84) reference genome. All the other data are available within the article and its Supplementary Information. Source data are provided with this paper. Uncropped blots are provided in Source data.

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

## Acknowledgements

This work was supported by grants from The Ministry of Science and Technology of China (2020YFC2002800) to F.Y. and Y.S., Beijing Natural Science Foundation (2019-A08) to L.C., Natural Science Foundation of China (81401845, 81471798, 82030035, 31620103904) to Y.Z., L.C. and Y.S., and Peak Disciplines (Type IV) of institutions of Higher Learning in Shanghai to Y.S.

## Author contributions

Y.Z. designed the research, conducted most of the experiments, analyzed data and wrote the paper. S.Li helped with cell culture. L.L., Y.K. and T.S. performed mouse surgery and intra-articular injection. S.Li, Q.W. and L.L. helped with collecting animal samples and data analysis. K.G. provided clinical samples. J.L. analyzed single-cell data. Y.T. was responsible for mice maintenance. S.Liu and R.S. contributed to molecular cloning and western blot. C.W. conducted FISH. J.W. performed single-cell platform. P.J., F.Y., W.Z. and X.Z. provided expertise and helped with experiments design. Y.S. designed the scRNA-seq and revised the paper. L.C. designed the project, supervised the study and revised the paper. All authors agreed on the final version of the paper.

## Competing interests

The authors declare no competing interests.

**Additional information**

