## [Peer Review File · Nature Communications]

Reviewers' Comments:

Reviewer #1:

Remarks to the Author:

The authors identified miR-17, which belongs to the miR-17~92 cluster, as a putative therapeutic target for OA. The miR-17~92 cluster is a critical for skeletal formation, and it is interesting that miR-17 is involved in the maintenance of cartilage homeostasis. However, the authors have insufficiently mention why miR-17 was selected from the cluster and the effects of other miRNAs on maintaining cartilage homeostasis by rescue experiments using each miRNA on cKO. The authors proved the function of miR-17 by reversing miR-17 to the cKO of the cluster, but in order to obtain more robust data, the authors should perform similar experiments using single miR-17 KO or miR-17 antogomir.

In addition, they performed scRNA-seq analysis, but the background and significance of the experiment is insufficiently explained. It should be examined and/or mentioned how the decrease in C1 and C2 percentages contributes to the increase in cartilage thickness and the subsequent increase in the expression of catabolic factors.

The expression changes of anabolic genes should be added in Fig. 2.

The statistical testing types should be added in Figure legend.

Reviewer #2:

Remarks to the Author:

The manuscript from Zhang et al describes the contribution of miR-17 to cartilage homeostasis in response to DMM-induced surgery, aging and in transgenic mice. The authors show that in DMM and aged mice, there is a reduction in the expression of miR-17 in articular chondrocytes of mice. Furthermore, intra-articular injectin of miR-17 agomir attenuate cartilage degeneration of the tibial plateau in the DMM, aging and miR-17~92 conditional knockout mice. Using in silico analyses, luciferase assays and western blotting, the authors identified MMP-13, ADAMTS5, NOS2 and MMP3 as direct targets of miR-17. The authors confirmed that MMP-13 and NOS2 were regulated in vitro models by injection of miR-17 agomir. The authors next show that GDF5 was able to increase miR-17 expression, rescue DMM-induced OA in a miR-17 dependent manner. Furthermore, cartilage modification in cKO was also modified by miR-17, but independently of identified target genes. Using scRNAseq, the authors identified three chondrocyte identities which were each modified by knockout of miR-17~92 cKO and rescued by miR-17. Finally, in human tissues, miR-17 expression was also modified in OA cartilage compared to normal.

Overall, the manuscript is very well written and provides an interesting mechanism for regulating chondrocyte phenotype and cartilage homeostasis related to miR-17 gene targets and function. However, there is some disconnect between what the authors show in the miR-17~92 knockout mice and other OA models, in addition to scRNAseq data, particularly in relation to the selected direct targets of miR-17. It is recommended that the authors re-evaluate their scRNAseq data taking into consideration genes predicted to be modified by miR-17 directly to identify direct targets modulated by miR-17 agomir and potentially confirm these by IHC using section from their cKO/agomir treated animals. Luciferase studies could also be completed using targets identified and follow-up with qPCR using samples already collected. Specific questions, comments and clarifications requested are indicated below.

- 1) The statistics used for comparisons should be indicated for each panel in each figure legend. It is unclear what tests were done for each analysis. Although the statistics section in the methods is well described, it would be helpful if the authors could include specific information in each legend to inform the reader what comparison tests were done and if they were parametric or non-parametric.
- 2) For figure S1B, is the cloudy area of fluorescence consistent with the agomir in the matrix? Was the same fluorescence excitation power used for these images as this type of cloudiness can occur under longer exposures and higher power.
- 3) A critical control that is missing from the manuscript is the effect of the agomir on sham or WT

mice. Was there any effect on these mice of the agomir? In contrast, was there an effect of antagomir on these mice? These experiments would be of interest to confirm the direct effect of endogenous mir-17 on cartilage homeostasis.

4) These selection of genes targeted by miR-17 appears to be very selective rather than allowing the data to dictate the targets to follow up on. Since miR-17 was detected in all zones, particularly in two separate superficial zone cell clusters, how does miR-17 maintain cell phenotype?

Specifically, in figure 5/6, the authors indicate that the MMP13, ADAMTS5, NOS2 and MMP3 are not modified in the miR-17~92 cKO mice, suggesting that other genes are likely to play significant roles in maintaining cartilage homeostasis in animal models of OA. It is recommended that the authors evaluate what targets may be directly modified by miR-17 using their scRNAseq data by identifying genes changed by in cKO and cKO with miR-17 agomiR compared to normal cells and cross-referencing them to predicted targets. This evaluation could give a more consistent set of genes that could be followed up in the other OA models investigated.

5) When the authors specifically talk about "chondrocytes" (like on Pg 7 line 16), it would be best to indicate the source of these cells (specifically from mouse or human origins) as both are used in the study.

6) In figure 2C, can the authors indicate what the level of the sham control is (instead of making it relative to sham, show the level of sham in the graph by indicating the delta-CT values for all mice)? This would specifically indicate how much of a change the DMM model induces these genes and allows readers to see the error in the baseline mice (for increased transparency).

7) ADAMTS5 IHC as well as NITEGE and C1,2C staining would be helpful to show changes in cartilage and aggrecan breakdown. Why did the authors use NOS2 as a marker as opposed to a catabolic enzyme/matrix marker for IHC follow-up?

8) In figure S3B, the colours of the bars here are too similar and hard to differentiate between those with and without IL-1.

9) In figure 3A, can the authors comment on if the level of mir-17 the same 4 weeks after GDF5 treatment as compared to 3 days? Also, is GDF5 expression altered in the models in this study, particularly the DMM, aging and cKO models?

10) Can the authors indicate how collagen rarefaction was measured/determined? There does not seem to be a figure or data corresponding to this statement on pg 11 line 6.

11) The authors indicate in the discussion that the miR-17~92 cluster has 6 miRNAs that target > 30 genes. What genes are specific to miR-17 in this cluster, if any? This may be important for selection of why miR-17 can rescue the models of OA in this study.

Reviewer #3:

Remarks to the Author:

1. Title should be more reflective of findings
2. Figure 1, panel A (and other figures with IF images): what is the difference between the last two rows of IF images? Label indicates red channel corresponds to miR-17 while blue corresponds to DAPI and bottom row shows both blue and red channels. Legend should be more descriptive.
3. Figure 1: Figure will be more streamlined if labels are consistent (i.e., placement on the same side of the figure). Additionally, authors should consider splitting panel A into at least 2 panels with their own identifier (A – IF and H&E, B – quantifications)
4. Figure 1 and results: Is there any difference in cell quantification outcomes if layer (i.e. superficial or middle) is taken into account?
5. Figure 1 and results: Are there any differences in OARSI scores when metric is split by quadrant? (for example, lateral vs medial)
6. Figure 1 and results: Was FISH used to look at miR-17/DAPI localization in agomir treated animals? Would agomir treatment affect the % of miR-17 cells?

7. Figure 1 legend: What is the age of the animals used for these experiments?
8. Page 6, line 17: Did you also use Cre- controls that were given tamoxifen? Cre- controls would be an important addition to validate the animal model.
9. Page 6, line 17: Could you explain why controls had an n=3 while the rest of the experiments seem to have at least an n=4?
10. Figure 1 (and others): Bottom of the legend stated that "dots represent BRs", can you specify what BRs are?
11. Are there any phenotype changes in the miR17-92 cKO mice? Mice seem to have a strong phenotype of cartilage degradation upon tamoxifen treatment, it would be important to validate that the phenotype is not present before tamoxifen treatment.
12. Figure 2: legend should be more descriptive of each of the panels that are present in the figure.
13. Page 9, line 10: Authors state that GDF5 was selected because this deficiency is the best replicated genetic risk factor for OA, can you provide supporting evidence for this and/or appropriate references?
14. Page 9, line 17: Authors state that GDF-5 restored miR-17 expression and number of positive cells, supporting figure only provides quantification of % cells, how do you quantify expression of miR-17 in these cells and can you provide that data?
15. Page 11, line 1: The authors state that even without DMM surgery they observe lack of proteoglycan staining in knee joints after tamoxifen injection. Given the important role of proteoglycans in maintaining tissue mechanical properties, what can you say about changes in joint loading leading to OA induction in this model (which could explain the changing OARSI scores without DMM surgery)?
16. Page 11, line 6: Authors state collagen rarefaction as a pathology outcome after tamoxifen induction, how is this quantified?
17. Figure 3, Panel A: Have the authors looked into cell morphology outcomes for the treatments included? Cells in vehicle control in the DMM group seem more elongated compared to cells in GDF-5 group which seem to be rounder.
18. Figure 3, panel B: sideways legend is confusing. Perhaps a table to explain how conditions are grouped would be easier to follow.
19. Figure 3 and results: It would be informative to have (either in the main text or the supplement) FISH images for Antagomir treatments
20. Page 12, line 15: It is surprising to not see Prg4 (lubricin) as one of the cell markers of the superficial zone, how does Prg4 expression look like in the dataset? Authors state in the discussion that there seemed to be two intermingled subpopulations of cells (one expressing Prg4) but this analysis is not obvious in the figures.

Response to the referee comments on NCOMMS-21-12281

We thank all three referees for kind reviewing of our manuscript. During the revision period, we have carefully studied all comments from the reviewers, and performed new experiments/analyses. Now we feel that we are able to address all of the reviewers' comments and made corresponding changes in substantially revised manuscript. The following are our point-by-point responses:

Reviewer #1 (Remarks to the Author):

The authors identified miR-17, which belongs to the miR-17~92 cluster, as a putative therapeutic target for OA. The miR-17~92 cluster is a critical for skeletal formation, and it is interesting that miR-17 is involved in the maintenance of cartilage homeostasis. However, the authors have insufficiently mention why miR-17 was selected from the cluster and the effects of other miRNAs on maintaining cartilage homeostasis by rescue experiments using each miRNA on cKO.

Response:

We would like to thank this reviewer for raising this important point. The *miR-17~92* cluster is a classic polycistronic miRNA gene cluster, in which four seed families and a total of 6 members including the miR-17 family (miR-17-5p and miR-20a-5p), the miR-18 family (miR-18a), the miR-19 family (miR-19a-3p and miR-19b-3p), and miR-92a-3p. In a published study by Han *et al.*, through serial mutations of each of the 4 seed family members within this cluster, only deletion of the miR-17 seed family resulted in vertebral transformations and shortening of the fifth mesophalanx, indicating that miR-17 has a profound role in controlling skeletal development in vertebrates (Han et al. *Nat Genet.* 2015;47(7):766-775). Since cartilage and skeletal development are closely related, it is not far-fetched to speculate that miR-17 might also regulate cartilage formation/degradation. During revision of the manuscript, we performed additional experiments further addressing the relevance of the other seed families in response to reviewer's question. We performed quantitative PCR using chondrocyte samples and discovered that the expression levels of miR-17 were substantially higher than that of the rest five members belonging to the *miR-17~92* cluster (see below, and Supplementary Fig. 1i of revised manuscript). Based on expression levels, we excluded miR-18a and miR-19a-3p for further consideration due to their very low expression in chondrocytes.

Supplementary Fig. 1 i Relative expression level of miRNA members belonging to miR-17-92 cluster in normal articular cartilage. $n = 3$ biologically independent samples.

Furthermore, as suggested by the reviewer, we performed rescue experiments by injecting miR-17, miR-19b-3p (miR-19) and miR-92a-3p (miR-92), respectively, into knee joints of *miR-17~92* cluster cKO mice. The result indicated that miR-19 did not elicit significant cartilage protection, while miR-92 demonstrated certain degrees of protection, and miR-17 appeared to be even more potent than miR-92 (see below, and Supplementary Fig.1j of revised manuscript). In agreement with the *in vivo* results, miR-19 could only suppress 2 out of the 4 catabolic factors, ADAMTS5 and NOS2, while miR-17 and miR-92 could regulate all 4 (MMP3, MMP13, ADAMTS5 and NOS2) (Supplementary Fig. 5 of revised manuscript). Since miR-92 has already been reported to inhibit cartilage catabolic factors in cultured chondrocytes (Mao et al. *Cell Physiol Biochem.* 2017;44,38-52), we obviously would choose to investigate miR-17 for novelty. In addition, miR-20 shares the same seed sequence as miR-17, and is expressed at much lower levels than miR-17, we decided to focus on miR-17 for this study. Although we have demonstrated critical roles of miR-17 in regulating cartilage homeostasis, we do not exclude the possibility that miR-92, miR-20, or even miR-19 also may somehow participate in the regulation. We have added these new data and the reasoning for choosing miR-17 in our revised manuscript (Page 6-7).

Supplementary Fig. 1 j Representative images of safranin O/fast green staining and OARSI grade from miR-17~92 cKO mice at 4 weeks after DMM surgery. Agomir-NC, agomir-17,

agomir-19 or agomir-92 (2 nmol/injection) were injected weekly for 3 times beginning at 1 week after surgery. $n = 6$ mice per group. $*P < 0.05$, $**P < 0.01$, $***P < 0.001$.

The authors proved the function of miR-17 by reversing miR-17 to the cKO of the cluster, but in order to obtain more robust data, the authors should perform similar experiments using single miR-17 KO or miR-17 antagomir.

Response:

We agreed with the reviewer that we should perform loss-of-function studies of single miR-17 to investigate its role in maintaining cartilage homeostasis and protection against cartilage degradation during OA. As knockout of single miR-17 could not be achieved in the requested revision time, we performed intra-articular injection of antagomir-17 in sham- or DMM-operated mice as also suggested by the reviewer.

Our results indicated that 3 days after injection of antagomir-17, at dose of 3 nmol, endogenous miR-17 was no longer detected in joint cartilage (Supplementary Fig. 1d of revised manuscript). In DMM joints, antagomir-17 resulted in remarkable exacerbation of cartilage destruction, supporting the protective role of miR-17. However, the severity was still less than that in cKO mice after DMM (see below, and Fig. 1c, e of revised manuscript). Knockdown of endogenous miR-17 by injecting antagomir-17, unlike deletion of the whole miR-17~92 cluster, failed to trigger a spontaneous OA pathology (Fig. 1c). This is likely because other members of the miRNA, such as miR-92 and miR-20, might elicit compensatory effects.

Fig. 1 c Safranin-O staining and scoring of OARSI grade in mice subjected to sham operation or DMM surgery and weekly intra-articular injection of antagomir-NC or antagomir-17 (3nmol) for three times, starting from 1 week after surgery ($n = 6-8$ mice per group). **e** Representative images of safranin O/fast green staining and OARSI grade from miR-17~92 cKO mice and their control littermates (miR-17~92^{fl/fl}) at 4 weeks after DMM surgery. $n = 6$ mice per group. The cKO mice and their controls were injected with tamoxifen at the age of 7-week-old and performed surgeries at the age of 8-week-old. NS, not significant, $*P < 0.05$, $**P < 0.01$, and $***P < 0.001$.

In addition, they performed scRNA-seq analysis, but the background and significance of the experiment is insufficiently explained. It should be examined and/or mentioned how the decrease in C1 and C2 percentages contributes to the increase in cartilage thickness and the subsequent increase in the expression of catabolic factors.

Response:

We are sorry about the insufficient explanation of the background and significance of the scRNA seq experiments. It has been well recognized that the joint cartilage is composed of four distinct layers and chondrocytes showed substantial heterogeneity in cellular morphology, arrangement, and matrix production. However, little is known about the difference at the whole transcriptomic levels among the chondrocyte subsets. Additionally, changes in gene transcription of each chondrocyte subset under OA pathology have not been fully elucidated. Here, using scRNA-seq analysis, we identified three distinct subpopulations of chondrocytes which resided in superficial (cluster C1, C2), middle (cluster C3) layers with specific transcriptomic features. Unexpectedly, we found robust expression of genes regulating not only anabolism but catabolism in healthy superficial chondrocytes, indicating active matrix turnover in normal joint cartilage. These three cell populations underwent dysregulation of both anabolism and catabolism in degenerated OA cartilage, which could be restored with miR-17 supplementation. These data helped us to know more deeply about biological activities of each chondrocyte subsets under physiology circumstances, and to explore the mechanism underlying cartilage degeneration in *miR-17~92* cKO mice. We have added the background and significance of the scRNA-seq analysis in our revised manuscript (Page 9, Line 217-223; Page 14, Line 359-369).

We are sorry about not mentioning the interrelationship between decreases of superficial chondrocytes and increases of cartilage thickness in our original manuscript. The superficial zone was found to be a structurally important element, which greatly limits swelling of the entire cartilage layer (Setton et al. *J Biomech Eng.* 1998;120(3):355-361. Brown et al. *J Anat.* 2021, doi:10.1111/joa.13527) by maintaining fluid pressurization within the tissue (Patel et al. *Adv healthc Mater.* 2021;10(10):e2100315; Owen et al. *Biomech Model Mechanobiol.* 2006; 5(2-3):102-110). Thus, decreases in numbers of superficial chondrocytes and subsequent degeneration of superficial zone contributed to cartilage edema, leading to increases

of cartilage thickness. We added this explanation in the discussion session of the manuscript (Page 14, Line 347-351).

In addition, the reviewer raised the question about whether decreased number of superficial chondrocytes would result in increases in the expression of catabolic factors. We believe this point might be mis-taken by the reviewer, because our data did not show knockout of the *miR-17~92* cluster led to increased expression of catabolic factors (Fig.S5A in the original manuscript and Supplementary Fig. 8 of the revised manuscript).

The expression changes of anabolic genes should be added in Fig. 2.

Response:

Following this reviewer's comments, we added the western blot analysis of anabolic factors in chondrocytes in Fig. 2b of the revised manuscript. The results showed that miR-17 overexpression in IL-1 β stimulated chondrocytes did not increase the expression of anabolic factors SOX9 and COL2A1.

The statistical testing types should be added in Figure legend.

Response:

Following this reviewer's suggestions, we added the statistical testing types in Figure legends of revised manuscript.

Reviewer #2 (Remarks to the Author):

The manuscript from Zhang et al describes the contribution of miR-17 to cartilage homeostasis in response to DMM-induced surgery, aging and in transgenic mice. The authors show that in DMM and aged mice, there is a reduction in the expression of miR-17 in articular chondrocytes of mice. Furthermore, intra-articular injection of miR-17 agomir attenuate cartilage degeneration of the tibial plateau in the DMM, aging and miR-17~92 conditional knockout mice. Using in silico analyses, luciferase assays and western blotting, the authors identified MMP-13, ADAMTS5, NOS2 and MMP3 as direct targets of miR-17. The authors confirmed that MMP-13 and NOS2 were regulated in vitro models by injection of miR-17 agomir. The authors next show that GDF5 was able to increase mir-17 expression, rescue DMM-induced OA in a miR-17 dependent manner. Furthermore, cartilage modification in cKO was also modified by miR-17, but independently of identified target genes. Using scRNAseq,

the authors identified three chondrocyte identities which were each modified by knockout of miR-17~92 cKO and rescued by miR-17. Finally, in human tissues, miR-17 expression was also modified in OA cartilage compared to normal.

Overall, the manuscript is very well written and provides an interesting mechanism for regulating chondrocyte phenotype and cartilage homeostasis related to miR-17 gene targets and function. However, there is some disconnect between what the authors show in the miR-17~92 knockout mice and other OA models, in addition to scRNAseq data, particularly in relation to the selected direct targets of miR-17. It is recommended that the authors re-evaluate their scRNAseq data taking into consideration genes predicted to be modified by miR-17 directly to identify direct targets modulated by miR-17 agomir and potentially confirm these by IHC using section from their cKO/agomir treated animals. Luciferase studies could also be completed using targets identified and follow-up with qPCR using samples already collected. Specific questions, comments and clarifications requested are indicated below.

Our response: We would like to thank the reviewer for his/her overall positivity towards our study. In the following, we provide detailed responses to the instructive comments/suggestions from the reviewer.

1) The statistics used for comparisons should be indicated for each panel in each figure legend. It is unclear what tests were done for each analysis. Although the statistics section in the methods is well described, it would be helpful if the authors could include specific information in each legend to inform the reader what comparison tests were done and if they were parametric or non-parametric.

Response:

Following this reviewer's comments, we added the statistical testing types in Figure legends of the revised manuscript.

2) For figure S1B, is the cloudy area of fluorescence consistent with the agomir in the matrix? Was the same fluorescence excitation power used for these images as this type of cloudiness can occur under longer exposures and higher power.

Response:

Thanks for the careful scrutiny. The original Figure S1B is now presented as Supplementary Fig. 1f in revised manuscript. We think the fluorescence is consistent with the distribution of the agomir in the extracellular matrix. The “cloudy area” is not a result of longer exposure or higher power of light as the reviewer wondered, because the same excitation power was used for all three groups of images. The position of the “cloudy area” might be at the deep and/or calcified zones and the “cloudy signals” might represent extracellular cholesterol-modified agomir clusters failed to enter the cells and got trapped in the extracellular matrix of that region. With time, these “cloudy signals” gradually disappeared, as expected.

3) A critical control that is missing from the manuscript is the effect of the agomir on sham or WT mice. Was there any effect on these mice of the agomir? In contrast, was there an effect of antagomir on these mice? These experiments would be of interest to confirm the direct effect of endogenous mir-17 on cartilage homeostasis.

Response:

To investigate the effect of miR-17 overexpression on joint cartilage of sham-operated WT mice, we injected agomir-17 into knee joint of mice as recommended by the reviewer. We found that administration of agomir-17 had no influence on the cartilage structure in sham-operated WT mice (Fig. 1d). This result indicates that supplement of miR-17 elicits no further effect on joint cartilage when endogenous miR-17 maintains at the physiological level. The effect of antagomir-17 on cartilage was already answered above in our responses to reviewer # 1.

4) These selection of genes targeted by miR-17 appears to be very selective rather than allowing the data to dictate the targets to follow up on. Since miR-17 was detected in all zones, particularly in two separate superficial zone cell clusters, how does miR-17 maintain cell phenotype? Specifically, in figure 5/6, the authors indicate that the MMP13, ADAMTS5, NOS2 and MMP3 are not modified in the miR-17~92 cKO mice, suggesting that other genes are likely to play significant roles in maintaining cartilage homeostasis in animal models of OA. It is recommended that the authors evaluate what targets may be directly modified by miR-17 using their scRNAseq data by identifying genes changed by in cKO and cKO with miR-17 agomiR compared to normal cells and cross-referencing them to predicted targets.

This evaluation could give a more consistent set of genes that could be followed up in the other OA models investigated.

Response:

To predict the direct targets of miR-17, microRNA target prediction programs including miRanda and TargetScan, were used to scan the 3'-UTRs of mRNAs. Since microRNAs modulate gene expression by suppressing mRNA translation into proteins, we postulated that miR-17 protected against cartilage destruction by inhibiting catabolic factors. In total, nine potential targets relevant to catabolism in OA cartilage were selected, which includes *Epas1*, *Mmp2*, *Mmp3*, *Mmp13*, *Mmp24*, *Adamts5*, *Adamts12*, *Adam9* and *Nos2*. After confirming that *Epas1* was not a direct target for miR-17 (Fig. 2a), we decided to select 3'-UTRs of *Mmp3*, *Mmp13*, *Adamts5* and *Nos2* for further characterization (Supplementary Fig. 3), because these 4 factors are known to be major catabolic factors in OA. Luciferase reporter assays verified that miR-17 suppressed the expression of all 4 genes through 3'UTR seed recognition sequences. Following the reviewer's comments, we revised the manuscript to show how the targets were selected, which could be found in Page 6-7, Line 157-163.

Although scRNA-seq is quite powerful for identification of new cell subtypes, the current 10X-genomics-based scRNA-seq technology only has an average mRNA capture rate around 15-20%, which means the majority (80-85%) of the mRNA expressed in a cell will not be captured or sequenced. Thus, the current scRNA-seq results were often not deep enough to allow for construction of full transcription regulatory net-works. As some of the critical transcription factors are not expressed at very high levels, they could easily be missed out in scRNA-seq. Moreover, mRNA levels and protein levels may not always change in the same direction, making functional predictions from only mRNA studies challenging.

That being said, we still tried very hard to follow this reviewer's suggestion to search for additional critical miR-17 targets to explain its role in regulating cartilage homeostasis. We reviewed our scRNA-seq data, and found that genes for collagen-modifying enzymes were abundantly increased in cKO chondrocyte populations, including those of collagen prolyl hydroxylase (*P3h3*), lysine hydroxylase (*Plod2*) and lysyl oxidase (*Lox*), some of which were inhibited following agomir-17 injection (Fig. 6e). Simultaneously, expressions of genes for collagen (*Col3a1* in C1; *Col1a1* and *Col22a1* in C2; *Col2a1* in C3) were reduced (Supplementary Fig. 8). Reduced collagen synthesis and increased collagen modifications were also reported

previously in growth plate cartilage of *Phd2^{chon-}* mice with accumulation of HIF-1 α , allowing us to speculate that HIF-1 α might play critical roles in dysregulation of collagen turnover in miR-17~92 cKO mice (Stegen et al. *Nature*. 2019;565(7740):511-515). We thus examined expression levels of HIF-1 α in joint cartilage by western blot in wild type and cKO mice and found that HIF-1 α protein levels were elevated in cKO cartilage, along with decreased expression of SOX9, COL2A1, and MMP2. With administration of agomir-17, changes in the expression of HIF-1 α , SOX9, COL2A1, and MMP2 were reversed (see below, Fig. 6f). In cultured chondrocytes treated with PHD2 inhibitor (IOX2 at 20 μ M) for 72 h, HIF-1 α was elevated and levels of *P3h3*, *Plod2* and *Lox* were increased. While the expression of SOX9 and MMP2 were decreased. Treatment with combination of miR-17 inhibitor and IOX2 caused stronger expression of HIF-1 α and collagen-modification enzymes, further decreases of SOX9 and MMP2. In contrast, overexpression of miR-17 resulted in significant decreases of HIF-1 α , along with increased expression of SOX9, MMP2 and decreased expression of collagen-modification enzymes (Fig. 6g-i). However, we did not observe significant changes in COL2A1 protein expression upon IOX2 and miR-17 inhibitor treatment, probably because of increased collagen modifications, protecting COL2A1 from degradation. This is slightly different from the *in vivo* situation, potentially due to a lack of mechanical loading in culture conditions. Given that the target site of miR-17 in 3'UTR of HIF-1 α has been well characterized in previous report using miRNA-reporter assays (Nunes et al. *PNAS*. 2015;112(12):3770-3775), we conclude that miR-17 maintains cartilage homeostasis through directly targeting HIF-1 α , by which collagen turnover was sustained in balance under physiological circumstances.

In summary, miR-17 is a regulator for HIF-1 α in articular chondrocytes, and homeostatic balance between catabolism and anabolism was maintained under physiological HIF-1 α expression. In DMM-induced OA, pathological catabolic factors targeted by miR-17 were increased. In *miR-17~92* cKO mice, miR-17 deficiency caused pathological increase of HIF-1 α , thereby decreasing the physiological catabolism and anabolism in chondrocytes. Supplementation with exogenous miR-17 or inducing endogenous miR-17 by GDF-5 treatment prevented OA and restored cartilage homeostasis (see below, Fig. 7). By clarifying the role of miR-17/ HIF-1 α axis in cartilage homeostasis, we believe that the integrity and strengthen of our study was enhanced, avoiding disconnect between cKO mouse and DMM model. We added these new results and relevant discussion in revised manuscript (Page 11,12, and 15).

Fig. 6 e, Violin plots showing changes in levels of collagen modifying genes in clusters C1, C2, and C3.

f, Protein levels of HIF-1α, SOX9, COL2A1 and MMP2 in articular cartilage of control, miR-17~92 cKO and cKO + agomir-17 mice at 2 weeks after tamoxifen injections were determined by Western blotting followed by densitometry analysis. Blots are representative of three independent experiments.

g, Mouse articular chondrocytes were transfected with miR-17 inhibitor (100 nM) or inhibitor NC or miR-17 mimic (50 nM) or mimic NC and treated with or without IOX2 (20μM) for 72 h. The protein levels of HIF-1α and SOX9 were determined by Western blotting followed by densitometry analysis. Blots are representative of three independent experiments.

h,i, Mouse articular chondrocytes were transfected with miR-17 inhibitor (100 nM) or inhibitor NC (**h**) or miR-17 mimic (50 nM) or mimic NC (**i**) and treated with or without IOX2 (20μM) for 72 h. The gene expression of P3h3, Plod2 and Lox was determined by qRT-PCR.

Data are presented as means ± s.e.m. * $P < 0.05$ versus control group, ° $P < 0.05$ versus cKO group in **f**. * $P < 0.05$ versus none-IOX2 group, ° $P < 0.05$ versus IOX2 with inhibitor/mimic NC group in **g**. * $P < 0.05$, ** $P < 0.01$, and *** $P < 0.001$ in **h, i**.

Fig. 7 Model depicting the mechanisms of miR-17 in maintaining cartilage homeostasis and prevention of OA.

5) When the authors specifically talk about “chondrocytes” (like on Pg 7 line 16), it would be best to indicate the source of these cells (specifically from mouse or human origins) as both are used in the study.

Response:

Thanks for reviewer’s careful scrutiny. We have indicated the origin of chondrocytes from mouse or human in revised manuscript.

6) In figure 2C, can the authors indicate what the level of the sham control is (instead of making it relative to sham, show the level of sham in the graph by indicating the delta-CT values for all mice)? This would specifically indicate how much of a change the DMM model induces these genes and allows readers to see the error in the baseline mice (for increased transparency).

Response:

Thanks for this suggestion. Now we showed levels of gene expression in sham control for all mice in revised graph for Fig.2c to visualize variations in the baselines.

7) ADAMTS5 IHC as well as NITEGE and C1,2C staining would be helpful to show changes in cartilage and aggrecan breakdown. Why did the authors use NOS2 as a marker as opposed to a catabolic enzyme/matrix marker for IHC follow-up?

Response:

As suggested by the reviewer, it would be helpful to detect ADAMTS5 and breakdown products of collagen and aggrecan in OA cartilage. While MMPs mainly degrade collagen and aggrecan, ADAMTS5 is an even more important enzyme for aggrecan degradation (Glasson et al, *Nature*. 2005;434(7033):644-648). Following reviewer's suggestion, we performed ADAMTS5 IHC and found that ADAMTS5 was upregulated in DMM cartilage and could also be reduced by miR-17 overexpression or GDF5 injection. We added these new results in Fig. 2d, f and Fig. 3d of the revised manuscript.

To directly detect matrix destruction, it is of importance to examine the degradation products such as the aggrecan NITEGE epitope, and the collagen C1,2C (Col2 $\frac{3}{4}$ C_{short}) neoepitope in OA cartilage. Antibodies against these epitopes could be purchased from a Canadian company, IBEX Pharmaceuticals Inc. (NITEGE epitope, product number: 50-1006; C1,2C (Col2 $\frac{3}{4}$ C_{short}), product number: 50-1035) (Lin et al, *Nat Med*. 2009;15(12):1421-1425). Unfortunately, we have been unable to find reliable distributors for these antibodies in China, particularly due to the COVID-19 pandemic. We found another Aggrecan Neo antibody (Invitrogen, PA1-1746), but was out of stock at the moment. We hope we can acquire these antibodies in our future studies. That being said, we believe that without these additional analyses, the main novel conclusions of the study still hold.

Nitric oxide (NO) acts as one of the major mediators that prompt the production of proinflammatory cytokines and catabolic factors in osteoarthritis. Increased production of NO in affected joints promote cartilage damage by upregulating levels of MMPs and inducing death of chondrocytes. As NO synthesis is catalyzed by the inducible NO synthase (iNOS, NOS2) enzyme, NOS2 inhibition serves as an attractive therapeutic target to prevent NO release (Leonidou et al. *Expert Opin Ther Targets*. 2018; 22(4):299-318.). Moreover, we identified seed sequence of miR-17 in 3'UTR of NOS2 mRNA, which was further evidenced by luciferase assay, we

therefore also included NOS2 as an additional critical target of miR-17 in our study.

8) In figure S3B, the colours of the bars here are too similar and hard to differentiate between those with and without IL-1.

Response:

Thanks to this suggestion. We have changed the colors of the bars to make them easier to distinguish from each other. Now the figure was numbered as Supplementary Fig. 6b in revised manuscript.

9) In figure 3A, can the authors comment on if the level of mir-17 the same 4 weeks after GDF5 treatment as compared to 3 days? Also, is GDF5 expression altered in the models in this study, particularly the DMM, aging and cKO models?

Response:

It is a good question concerning the long-term effect of repeated GDF-5 injection on miR-17 expression. We performed additional experiments by injecting GDF-5 once a week for four times and determined expression levels of miR-17 at days 14, and 28 days after the first GDF-5 or vehicle injection. It was shown that repeated GDF-5 injections could induce higher and more sustained levels of miR-17 expression (Supplementary Fig. 6d in revised manuscript), perhaps due to gradually improved cartilage micro-environment. We included these results in our revised manuscript (Page 8, Line 194-196).

Regarding GDF-5 expression in mouse models, it is well-known that GDF5 plays critical roles in chondrogenesis of fetal limbs. However, in adult healthy cartilage expression of GDF5 was diminished. In line with this, our scRNA seq data showed that *Gdf5* mRNA levels were extremely low in chondrocytes of adult mice (Feature plot of *Gdf5* in chondrocytes from control, cKO, and cKO+agomir-17 groups is shown below). Recent studies further demonstrated that *Gdf5* expression was transiently upregulated in cartilage at 2 weeks post-DMM, as well as in newly regenerated cartilage after injury (Kania et al. *Sci Rep.* 2020;10:157), suggesting that re-expression of GDF-5 in adult cartilage at early stage of injury. However, our scRNA-seq analysis showed that enrichment of *Gdf5* remained at extremely low levels in cKO chondrocytes and did not increase upon miR-17 injection. In addition, expression of GDF-5 was not detected by IHC staining in aged cartilage where spontaneous cartilage degeneration occurs (see below). Together, these results indicate that cartilage degeneration induced either by deletion of the *miR-17~92*

cluster or aging, was unable to trigger GDF-5 re-expression as was with DMM injury at early phases. Because we only used GDF-5 as an inducer for miR-17, we have not included these results in our manuscript.

Figure a, Feature plot of *Gdf5* in chondrocytes from control, *cKO*, and *cKO*+agomir-17 groups. **b-d**, IHC staining of GDF-5 in joint sections of *cKO* mice (**b**), WT mice subjected to sham or DMM surgery (**c**) and WT mice of 3-month-old and 12-month-old (**d**).

10) Can the authors indicate how collagen rarefaction was measured/determined? There does not seem to be a figure or data corresponding to this statement on pg 11 line 6.

Response:

Thanks for review's carefulness. Collagen rarefaction should be observed by Picosirius red staining under polarized light microscopy, which was not used in our study. Therefore, the statement in our original manuscript was inaccurate or erroneous. We have since changed the phrase "collagen rarefaction" to "matrix splitting" (Page 8, Line 208), which means horizontal matrix separation within middle or deep cartilage layer (Meachim et al. *Arthritis Rheum.* 1978;21(6):669-74. Byers et al. *Ann Rheum Dis.* 1976;35(2):114-21).

11) The authors indicate in the discussion that the miR-17~92 cluster has 6 miRNAs that target > 30 genes. What genes are specific to miR-17 in this cluster, if any? This may be important for selection of why miR-17 can rescue the models of OA in this study.

Response:

Thanks for this good suggestion. In the review article by Bai *et al*, cited in our manuscript, amongst the 30 confirmed targets for the *miR-17-92* cluster, 20 genes were predicted to be targeted by miR-17. Among the 20 putative miR-17 targets, 11 are OA or cartilage-related (see table below), but none of which were related to catabolism in OA cartilage, probably because the review article does not deal with OA. In our current study, miR-17 was predicted to target nine potential genes relevant to catabolism in OA cartilage, which were *Epas1*, *Mmp2*, *Mmp3*, *Mmp13*, *Mmp24*, *Adamts5*, *Adamts12*, *Adam9* and *Nos2*. miR-19 was predicted to target *Adamts17* and *Adam12*. miR-92 was predicted to target *Mmp16*, *Adam10*, *Adam23*, and *Adamts9*. Since MMP3 and ADAMTS5 are the two most important enzymes for cartilage degradation and HIF-2 α (*Eaps1*) is a crucial upstream catabolic regulator, we firstly selected miR-17 in this study. Using luciferase reporter assays, pathological catabolic factors including *Mmp3*, *Mmp13*, *Adamts5*, and *Nos2* were identified as direct targets of miR-17.

As mentioned above, amongst the 11 targets relevant to OA, HIF-1 α has been confirmed as target for miR-17 seed family (Taguchi *et al. Cancer Res.* 2008; Nunes *et al. PNAS.* 2015). Thanks to this reviewer, when we examined HIF-1 α in cartilage of cKO mice, accumulation of protein was apparent, which should result in suppression of anabolism (Stegen *et al. Nature.* 2019;565(7740):511-515). Together, in revised manuscript, we demonstrated both catabolic and anabolic targets for miR-17, confirming its involvement in regulating cartilage homeostasis and prevention against OA.

Table The targets of miR-17~92 and miR-17 summarized from a review article (Bai et al. *Biomed. Res. Int.* **2019**, 9450240)

miRNA name	Number of targets	Target genes
miR-17~92 cluster	30	APP,CCND1,TBC1D2,E2F,MAPK9,LATS2,BCL2L11,IRF, JNK2,MYCN,PKD2,GAB1,RBL1,TSG101,P63,STAT3,TGF B,HIF1A,RBL2,P57,P27,SHH,CTGF,TSP1,PTEN,BMP2,P21,SOCS,FOG2,ASK1
miR-17	20	APP,CCND1,TBC1D2,E2F,MAPK9,BCL2L11,IRF,MYCN,PKD2,GAB1,RBL1,TSG101,STAT3,TGFBR2,HIF1A,RBL2,TSP1,PTEN,BMP2,P21
	11	(OA or cartilage-related) CCND1,MAPK9,IRF,RBL1,STAT3,TGFBR2,HIF1A,TSP1,PTEN,BMP2,P21

Reviewer #3 (Remarks to the Author):

1. Title should be more reflective of findings

Response:

We agree with the reviewer's suggestion. We have revised the title to 'Dual functions of microRNA-17 in maintaining cartilage homeostasis and protection against osteoarthritis'.

2. Figure 1, panel A (and other figures with IF images): what is the difference between the last two rows of IF images? Label indicates red channel corresponds to miR-17 while blue corresponds to DAPI and bottom row shows both blue and red channels. Legend should be more descriptive.

Response:

Thanks for this reviewer's comments. The IF images in panel A showed decreased expression of miR-17 with progression of OA after DMM surgery. The bottom rows were the merged pictures of miR-17 (red channel) and DAPI (blue channel), showing intracellular miR-17 expression. We have changed figure legend in our revised manuscript to be more descriptive.

3. Figure 1: Figure will be more streamlined if labels are consistent (i.e., placement on the same side of the figure). Additionally, authors should consider splitting panel A into at least 2 panels with their own identifier (A – IF and H&E, B – quantifications)

Response:

Thanks for the great suggestion, we have split panel A into 2 panels to make it clearer. Also, we have made efforts to revise all figure legends to make them more streamlined.

4. Figure 1 and results: Is there any difference in cell quantification outcomes if layer (i.e. superficial or middle) is taken into account?

Response:

This is a great suggestion, which links miR-17 distribution in cartilage layers to the phenotype of cKO mice. Following this reviewer's suggestion, we calculated miR-17 positive cells by FISH within superficial and middle layers, respectively, of mouse joint cartilage. In normal knee cartilage, the percentage of miR-17 positive cells in superficial or middle layer to all positive cells was 31.32% and 64.91%, respectively. Within the superficial layer, miR-17 positive cells composed about 70% of all DAPI⁺ cell, whereas in the middle layer, the number is about 65%. At 4 weeks after DMM surgery, the percentage of miR-17-positive cells dropped from 70.39% to 30.83% in the superficial layer, and from 65.75% to 36.1% in the middle layer (Supplementary Fig. 1b). Since cells in the superficial layer were the first to sense stress signals in joint cartilage after DMM, the dramatic drop of miR-17, particularly in the superficial layer, provided an important indicator and a cause for subsequent OA development. In addition, superficial chondrocytes also appeared to be more sensitive to *miR-17~92* deletion than middle layer chondrocytes, demonstrating the importance of miR-17 expression in superficial chondrocyte homeostasis. We included the above discussion in Page 14, Line 351-356.

5. Figure 1 and results: Are there any differences in OARSI scores when metric is split by quadrant? (for example, lateral vs medial)

Response:

The reviewer suggested that the joint section should be split by quadrant for scoring. This method is useful for coronal (frontal) sections, which shows both of medial and lateral joints. Surgical DMM resulted lesions primarily on the central weight-bearing

region, specifically the anterior-central portion of the medial tibial plateau and medial femoral condyles (Glasson et al, *Osteoarthritis Cartilage*. 2007;15(9):1061-1069). In our study, we used sagittal sections of medial part of joints, without sectioning the lateral joints. Moreover, scoring of the tibial plateaus was highly reproducible and more consistent than that of the femur due to the much thinner femoral cartilage (Glasson et al, *Osteoarthritis Cartilage*. 2010;18 Suppl 3:S17-23). Therefore, we scored the cartilage destruction from the medial tibial cartilage in DMM model.

6. Figure 1 and results: Was FISH used to look at miR-17/DAPI localization in agomir treated animals? Would agomir treatment affect the % of miR-17 cells?

Response:

We have performed FISH to look at miR-17 location in agomir-treated mice, and the results were shown in Supplementary Fig.1f in revised manuscript. These pictures showed dynamic distributions of injected agomir-17 at 3 and 5 days post-injection. At 3 days post-injection, most of the fluorescence-positive cells were distributed in middle cartilage. At 5 days post-injection, positive cells were distributed in deep and calcified cartilage, suggestive of penetration of agomir-17 into calcified layer of joint cartilage. The agomir treatment upregulated the percentage of miR-17 positive cells from 17.86% at day 0 to 57.61% and 50.28% at 3 and 5 days, respectively, after injection.

7. Figure 1 legend: What is the age of the animals used for these experiments?

Response:

In this study, we use adult mice at the age of 10 weeks for experiments. It has been reported that male mice at 8 to 12 weeks of age are recommended for DMM surgery (Dai et al, *Ann Rheum Dis*. 2017;76(7):1295-1303. Choi et al, *Nature*. 2019;566:254-258), because the male mice become sexually and skeletally mature at the age of 8 to 9 weeks.

For cKO mice, we injected tamoxifen at the age of 7 weeks, before mice reached adulthood, and performed surgery at the age of 8 weeks, when the mice reached sexual maturity. We choose mice at this age because it has been reported that efficacy of Cre-recombination declines in adult *Col2a1-CreER^T* transgenic mice beyond the age of 8 weeks (Nakamura et al, *Dev Dyn*. 2006;235(9):2603-2612). This information about ages of animals was included in the figure legends and methods of the revised manuscript.

8. Page 6, line 17: Did you also use Cre- controls that were given tamoxifen? Cre-controls would be an important addition to validate the animal model.

Response:

Thanks for this suggestion. We used the Cre- mice(*miR-17-92^{fl/fl}*) that were given tamoxifen as controls. We added the results about this control in Supplementary Fig. 1h, as suggested by the reviewer.

9. Page 6, line 17: Could you explain why controls had an n=3 while the rest of the experiments seem to have at least an n=4?

Response:

Thanks for reviewer's carefulness. After checking the data, in the original Figure 1D should have been n = 4. We made a mistake in stating n = 3. We have revised the data in supplementary information, which is now shown in Supplementary Fig. 1h.

10. Figure 1 (and others): Bottom of the legend stated that "dots represent BRs", can you specify what BRs are?

Response:

We are sorry that we missed the full name of BRs, which represents biological replicates. Now BRs have been changed to "biologically independent samples or individual mice" to avoid misunderstanding.

11. Are there any phenotype changes in the miR17-92 cKO mice? Mice seem to have a strong phenotype of cartilage degradation upon tamoxifen treatment, it would be important to validate that the phenotype is not present before tamoxifen treatment.

Response:

Thanks for this concern. We did examine the phenotype cKO mice before tamoxifen treatment. Joint cartilage was healthy and cartilage degradation was not observed before tamoxifen treatment in *miR-17-92* cKO mice. The safranin O staining and corresponding OARSI scores of joints from cKO mice and control mice before tamoxifen injection were shown below.

12. Figure 2: legend should be more descriptive of each of the panels that are present in the figure.

Response:

Thanks for this suggestion. We have revised the legend of Fig. 2 to describe each panel in a clearer way. The changes were highlighted in yellow in manuscript related files.

13. Page 9, line 10: Authors state that GDF5 was selected because this deficiency is the best replicated genetic risk factor for OA, can you provide supporting evidence for this and/or appropriate references?

Response:

We are sorry about missing the reference for this statement. We have added the reference (Sun et al. *Cell Prolif.* 2021;54(3):e12998) in Page 7, Line 184.

14. Page 9, line 17: Authors state that GDF-5 restored miR-17 expression and number of positive cells, supporting figure only provides quantification of % cells, how do you quantify expression of miR-17 in these cells and can you provide that data?

Response:

We have provided expression data of miR-17 under GDF-5 administration by qRT-PCR in Supplementary Fig. 6d. The data showed GDF-5 significantly upregulated miR-17 levels from 1 to 5 days after GDF-5 injection.

15. Page 11, line 1: The authors state that even without DMM surgery they observe lack of proteoglycan staining in knee joints after tamoxifen injection. Given the

important role of proteoglycans in maintaining tissue mechanical properties, what can you say about changes in joint loading leading to OA induction in this model (which could explain the changing OARSI scores without DMM surgery)?

Response:

The reviewer raised an important issue about the effect of proteoglycan loss on changes in mechanical loading that leads to development of OA in cKO mice. Proteoglycans play important roles in maintaining cartilage mechanical properties by attracting cations and water to provide tensile strength (Bank et al. *Biochem J.* 1998;330:345-351). With joint loading, the proteoglycan aggregates are compressed and distribute the force onto the joint surface, thereby reducing the pressure on the articular cartilage (Eckstein et al. *J Anat.* 2006;208:491-512). Degradation of proteoglycans resulted significant increases of intratissue strains in the articular cartilage, particularly in the superficial zone, thereby decreasing the load bearing capacity of the cartilage (Pastrama et al. *J Mech Behav Biomed Mater.* 2019;98:383-394.). Abnormal load-bearing capacity further triggers progressive loss of proteoglycan and loss of collagen organization and cross links from the superficial zone (Hosseini et al. *Osteoarthritis Cartilage.* 2013; 21:136-143. Arokoski et al. *Scand J Med Sci Sports.* 2000;10:186-198). Thus, the abnormal joint loading resulted from loss of proteoglycan in cKO mice leads to cartilage destruction characterized by matrix edema and splitting, as well as increased OARSI scores even without DMM-induced joint instability. We have added related discussion in the main text of revised manuscript Page 14, Line 399-371.

16. Page 11, line 6: Authors state collagen rarefaction as a pathology outcome after tamoxifen induction, how is this quantified?

Response:

Thanks for reviewer's carefulness. We have already responded to this point above. We have changed the phrase "collagen rarefaction" to "matrix splitting" (Page 8, Line 208).

17. Figure 3, Panel A: Have the authors looked into cell morphology outcomes for the treatments included? Cells in vehicle control in the DMM group seem more elongated compared to cells in GDF-5 group which seem to be rounder.

Response:

We are impressed by this reviewer's keen observations. By reviewing previous data, we found that after DMM, DAPI stained cell nuclei indeed became smaller and increased in number, suggesting that these cells might enter an injury-induced proliferative/de-differentiated state, which is characteristic of early OA (Pritzker et al. *Osteoarthritis Cartilage*. 2006;14,13-29). Interestingly GDF-5 treatment reversed this phenotype. We would consider investigating this issue in the future.

18. Figure 3, panel B: sideways legend is confusing. Perhaps a table to explain how conditions are grouped would be easier to follow.

Response:

Thanks for the great suggestion. We added a table (see below) at the top of the figure to explain treatment conditions in the figures. Now the figure is Fig. 3c.

				DMM	-	+	+	+
GDF-5	-	-	+	+
Antagomir	NC	NC	NC	17

19. Figure 3 and results: It would be informative to have (either in the main text or the supplement) FISH images for Antagomir treatments

Response:

We completely agree with the reviewer's comments. We have added the FISH images to confirm the effect of antagomir on miR-17 expression in Supplementary Fig. 1d of the revised manuscript.

20. Page 12, line 15: It is surprising to not see Prg4 (lubricin) as one of the cell markers of the superficial zone, how does Prg4 expression look like in the dataset? Authors state in the discussion that there seemed to be two intermingled subpopulations of cells (one expressing Prg4) but this analysis is not obvious in the figures.

Response:

We understand the concerns by the reviewer that *Prg4* was not selected as a marker for superficial chondrocytes in our original study. It has been well accepted that *Prg4* is expressed in superficial layer of joint cartilage and thus *Prg4* is recognized as a

biomarker for superficial chondrocytes. By scRNA-seq, we also observed that *Prg4* was expressed in superficial chondrocytes cluster C2, which was decreased in cKO mice and reversed by agomir-17. This was now presented in Supplementary Fig. 8. However, by scRNA-seq, *Prg4* mRNA was unexpectedly expressed also in middle zonal chondrocytes (The feature plot was shown below), perhaps not translated into proteins. That is why it was not included as a marker for cluster C2 in Fig. 5c of our initial manuscript. Following reviewer's comments, we added the followed figure (Supplementary Fig. 7a) and description (Page 9, Line 236-239) into our revised manuscript

Reviewers' Comments:

Reviewer #1:

Remarks to the Author:

The authors answered most of the concerns. However, it remains unclear how they selected miR17 from the miRNA17-92 cluster and 19 and 92 for additional validation.

It would be preferable to utilize TaqMan array for miRNAs or miRNA-seq for comparisons between different miRNAs

It is an essential part of the rationale for excluding some miRNAs from the analysis; thus, a detailed explanation is necessary.

Reviewer #2:

Remarks to the Author:

The authors have done an excellent job at revising their manuscript and have added a number of new experiments to their manuscript that helps to clarify many of the issues originally brought up in the previous review. Two critical but easily addressable points remain based on the new data provided.

1) With respect to OARSI scoring, the authors have detailed in their methods that "To grade OA severity, OARSI scoring system was performed by two observers blinded to the experimental groups to evaluate cartilage destruction in the medial femoral condyle and/or medial tibial plateau. For each joint, 3~5 sections were used for safranin O staining and the median score was representative for each biological samples." The scores provided only range from 0-6, suggesting that only one surface of the medial joint was actually scored, particularly based on the severity of some representative images provided where both the femur and tibial surfaces look to be highly degenerative. Can the authors clarify? OARSI recommends that the scoring either be provided for each joint surface individually, or use a sum score for all joint surfaces.

2) The conclusions based on HIF1A are not conclusive and thus the language associated with these results should be modified. Although there appears to be a strong regulation of HIF1A by mir-17, and that other genes are modified consistent with its changes in expression, these could be unrelated events. The mechanism provided by the authors is possible, but not conclusive based on the data provided. Please revise.

Reviewer #3:

Remarks to the Author:

Manuscript: NCOMMS-21-12281

Thank you for the opportunity to review the manuscript entitled "Dual functions of microRNA-17 in maintaining cartilage homeostasis and protection against osteoarthritis." The authors have adequately addressed most of the reviewers' comments since the initial submission. The manuscript has a few additional weaknesses that should be addressed before it is accepted for publication.

The following comments should be considered and addressed:

Major comments:

1. The authors identify 9 potential targets of miR-17 relevant to catabolism in OA cartilage but only select four factors for further characterization without adequate explanation of why these four factors were chosen. One of the 9 targets that is excluded from further characterization is Mmp2, which the authors later identify as a gene highly enriched in chondrocyte clusters but with decreased expression in cKO mice. Does Mmp2 get regulated through the same 3'-UTR region by miR-17 that the other 4 genes are regulated through?

Minor Comments

Line 156: Change To explored -> To explore

Line 166: Change eradicated -> eradicated

Line 238-239: Author states that cluster C, which represents the middle zonal chondrocytes, express Prg4a, a known superficial chondrocyte marker, but fails to adequately explain the reason for this unexpected finding. Why would Prg4 mRNA be present if the lubricin protein is not expressed?

Response to the referee comments on NCOMMS-21-12281A

We would like to thank all three referees for their kind reviews of our revised manuscript. We have carefully studied all comments from the reviewers, and performed new experiments/analyses. The following are our point-to-point responses:

Reviewer #1 (Remarks to the Author):

The authors answered most of the concerns. However, it remains unclear how they selected miR17 from the miRNA17-92 cluster and 19 and 92 for additional validation. It would be preferable to utilize TaqMan array for miRNAs or miRNA-seq for comparisons between different miRNAs

It is an essential part of the rationale for excluding some miRNAs from the analysis; thus, a detailed explanation is necessary.

Response: We first thank this reviewer for acknowledging our effort during the first round of revision. Regarding the rationale for selecting miR-17 from *miR-17~92* cluster in this study, we would like to use the following space to re-review the research path leading us to miR-17.

Although we did not perform TaqMan array or miRNA sequencing to screen for relevant miRNAs involved in OA, we did perform in-silico screening using 30 most-abundantly expressed miRNAs identified by Kobayashi et al. in chondrocytes (Kobayashi *et al. PNAS*. 2008.105(6):1949-1954), as well as miRNA target prediction programs (TargetScan and miRanda) to search for miRNAs predicted to simultaneously target MMP13 and ADAMTS5, which are the most important enzymes for degradation of collagen and aggrecan, respectively (Supplementary Fig.1a, and see below). Such a screen resulted in only 4 miRNA species, miR-17, miR-20a, miR-140-3p, and miR-27-3p. MiR-140-3p (Miyaki et al. *Genes & development*. 2010. 24(11):1173-1185) and miR-27-3p (Akhtar et al. *Arthritis Rheum*. 2010.62(5):1361-1371) have been reported to play essential roles in cartilage development and homeostasis. MiR-17 and miR-20a happened to be located in the *miR-17-92* cluster and shared the same targeting sequence. This drew our attention

to the *miR-17-92* cluster.

Secondly, it has been reported that growth differentiation factor 5 (GDF-5) modulates the phenotype of chondrocytes and plays key roles during joint morphogenesis. In addition, functional single-nucleotide polymorphisms (SNPs) linked to *Gdf5* deficiency are currently the most consistent risk factor for adult-onset osteoarthritis (OA) across human populations. Furthermore, skeletal defects resulted from dysfunction of *miR-17~92* cluster showed shortening of phalangeal elements and bony fusions of the joints, which are also observed with GDF5 deficiency, suggesting an inter-connection between the two (see introduction section, Page 4, Line 87-93). We thus investigated the possible relationship between GDF-5 and the *miR-17~92* cluster, and found that within the *miR-17~92* cluster (comprising miR-17, miR-18a, miR-19a-3p, miR-19b-3p, miR-20a and miR-92a-3p), miR-17 was the most highly expressed and strongly upregulated miRNA, in response to GDF-5. MiR-20a, which contained the same seed sequence as miR-17, was also significantly upregulated (Supplementary Fig. 1b in newly revised manuscript, and see below). These results suggested that miR-17 and/or miR-20a might mediate the effect of GDF-5 on chondrocytes. Since miR-17 shared the same seed sequence with miR-20a and had a higher expression level, we selected miR-17 first for further study. We also validated potential involvement of miR-92a-3p and miR-19b-3p, but not miR-18a, or miR-19a-3p, simply because miR-18a and miR-19a-3p were minimally expressed in chondrocytes and were not regulated by GDF-5 (Supplementary Fig. 1b in newly revised manuscript, and see below).

Currently we do not know whether miR-140 and miR-27 have overlapping or additional protective roles over miR-17/miR-20a during OA. This needs to be addressed in future studies.

We hope that these new pieces of information together with our responses during the first round fully addressed this reviewer's concerns about why we focused on studying miR-17, and we also accordingly revised the result sections in the newly revised manuscript (Line 101-115, Page 4-5).

Supplementary Fig. 1 a, In top 30 miRNAs expressed in chondrocytes, four miRNAs were predicted to target MMP13 and ADAMTS5 simultaneously by TargetScan and miRanda. The list of miRNAs predicted to target MMP13 or ADAMTS5 and the top 30 miRNAs in chondrocytes were provided in source data. **b**, Expression of all six members of the *miR-17-92* cluster, as determined by qRT-PCR analysis, in primary cultures of mouse articular chondrocytes treated with GDF-5 (0, 100 or 300 ng/mL) for 24 h. n = 3 per group. * $P < 0.05$, ** $P < 0.01$, and *** $P < 0.001$ compared with 0ng/mL GDF-5 (ANOVA with Bonferroni's test).

Reviewer #2 (Remarks to the Author):

The authors have done an excellent job at revising their manuscript and have added a number of new experiments to their manuscript that helps to clarify many of the issues originally brought up in the previous review. Two critical but easily addressable points remain based on the new data provided.

1) With respect to OARSI scoring, the authors have detailed in their methods that "To grade OA severity, OARSI scoring system was performed by two observers blinded to the experimental groups to evaluate cartilage destruction in the medial femoral condyle and/or medial tibial plateau. For each joint, 3~5 sections were used for safranin O staining and the median score was representative for each biological samples." The scores provided only range from 0-6, suggesting that only one surface of the medial joint was actually scored, particularly based on the severity of some representative images provided where both the femur and tibial surfaces look to be

highly degenerative. Can the authors clarify? OARSI recommends that the scoring either be provided for each joint surface individually, or use a sum score for all joint surfaces.

Response: Thanks for this reviewer's overall positivity towards our first round of revision, as well as the comment on OARSI scoring, which we would like to clarify here. In our study, only one surface of the medial joint, either tibial or femoral surface was scored depending on the condition. For example, surgical DMM resulted in lesions primarily on the central weight-bearing region, specifically the anterior-central portion of the medial tibial plateau and medial femoral condyles (Glasson et al, *Osteoarthritis Cartilage*. 2007;15(9):1061-1069). Moreover, scoring of the tibial plateaus was highly reproducible and more consistent than that of the femoral condyle due to femoral cartilage being much thinner (Glasson et al, *Osteoarthritis Cartilage*. 2010;18 Suppl 3:S17-23). Therefore, we only scored the cartilage destruction from the medial tibial cartilage in DMM model.

Exactly as the reviewer indicated, in some representative images both femoral and tibial surfaces looked to be highly degenerative, especially in cKO mice with DMM surgery or aged mice with DMM surgery. For consistency, we scored the cartilage destruction from the tibial cartilage in all models involving DMM.

In cKO mice without DMM, however, the cartilage lesion was observed primarily in femoral condyles. Therefore, we only scored the cartilage destruction from there.

We added the following description into the method section of the newly revised manuscript: *we scored the cartilage destruction from the medial tibial plateau in all DMM-involved models and medial femoral condyles in cKO mice without DMM surgery* (Line 502-503, Page 19).

2) The conclusions based on HIF1A are not conclusive and thus the language associated with these results should be modified. Although there appears to be a strong regulation of HIF1A by mir-17, and that other genes are modified consistent with its changes in expression, these could be unrelated events. The mechanism

provided by the authors is possible, but not conclusive based on the data provided. Please revise.

Response: Thanks for the reviewer's careful scrutiny. As a key regulator for chondrocyte survival and collagen production, HIF-1 α was negatively regulated by miR-17, which likely influenced the fate of chondrocytes, yet the reviewer is right that we did not provide enough evidence to prove causality. In newly revised manuscript, we toned down the claim by changing the statement 'miR-17 ... and maintains the physiological catabolic and anabolic balance by restricting HIF-1 α signaling' into 'miR-17 ... and maintains the physiological catabolic and anabolic balance potentially by restricting HIF-1 α signaling' in the abstract and other sections of newly revised manuscript.

Reviewer #3 (Remarks to the Author):

Thank you for the opportunity to review the manuscript entitled "Dual functions of microRNA-17 in maintaining cartilage homeostasis and protection against osteoarthritis." The authors have adequately addressed most of the reviewers' comments since the initial submission. The manuscript has a few additional weaknesses that should be addressed before it is accepted for publication.

The following comments should be considered and addressed:

Major comments:

1. The authors identify 9 potential targets of miR-17 relevant to catabolism in OA cartilage but only select four factors for further characterization without adequate explanation of why these four factors were chosen. One of the 9 targets that is excluded from further characterization is Mmp2, which the authors later identify as a gene highly enriched in chondrocyte clusters but with decreased expression in cKO mice. Does Mmp2 get regulated through the same 3'-UTR region by miR-17 that the other 4 genes are regulated through?

Response: Thanks for this reviewer's overall positivity towards our first round of

revision. We are sorry for inadequate presentation of why several catabolic genes were excluded for target validation. After the in-silico identification of 9 potential targets of miR-17 relevant to catabolism, we examined their expression patterns in OA chondrocytes both *in vitro* and *in vivo*. The results showed that *Epas1*, *Mmp3*, *Mmp13*, *Adamts5*, and *Nos2* were elevated significantly in IL-1 β -treated chondrocytes as well as DMM cartilage. In contrast, expression levels of *Mmp2* and *Adam9* were not changed, and *Mmp24* and *Adamtsl2* were non-detectable in OA chondrocytes (Supplementary Fig. 3a, b of newly revised manuscript, and see below). After confirming *Epas1* is not as a target of miR-17 (Fig.2a in the 1st round revised as well as this round of newly revised manuscript), catabolic factors including *Adamts5*, *Mmp3*, *Mmp13* and *Nos2* were chosen for further validation. We added this information in newly revised manuscript (Page 7, Line172-175).

In the last part of our study, we gained some evidence that miR-17 positively (likely indirectly) regulated MMP2 expression in normal chondrocyte potentially by restricting HIF-1 α signaling, and too prolonged HIF-1 α negatively regulates MMP2. Such double negative regulations eventually lead to positive regulation of MMP2 by miR-17. Thus, unlike *Mmp3/13*, *Mmp2* is likely not a direct target of miR-17 in chondrocytes, even though the potential target site was present in 3' UTR of *Mmp2*.

Supplementary Fig. 3 a, Mouse articular chondrocytes were treated with or without IL-1 β (5 ng/mL) for 24 h. The levels of catabolic factors were determined by qRT-PCR. n = 3-4 biologically independent samples per group.

b, qRT-PCR analysis of catabolic genes in knee cartilage of mice subjected to sham or DMM surgery for 4 weeks. n = 3 biologically independent samples per group. ND, no detection. NS, not significant, * P < 0.05, ** P < 0.01, and *** P < 0.001. Two-sided Student's

t-test.

Minor Comments

Line 156: Change To explored -> To explore

Line 166: Change eradicated -> eradicated

Response: We are sorry for these typing errors and have made corrections in newly revised manuscript.

Line 238-239: Author states that cluster C3, which represents the middle zonal chondrocytes, express Prg4a, a known superficial chondrocyte marker, but fails to adequately explain the reason for this unexpected finding. Why would Prg4 mRNA be present if the lubricin protein is not expressed?

Response: Thanks for this concern. In our data, expression levels of Prg4 in control mice of superficial C2 were high. C3 representing middle zonal chondrocytes also expressed certain levels of Prg4 mRNA (see below). A study (Chau et al. *PLOS ONE*. 2014. 9(7): e103061) also reported that Prg4 mRNA was expressed in the intermediate/deep zone of the articular cartilage 25-fold higher than the resting zone of growth plate cartilage (raw signal:1265 vs 51, table 4 in the Chau et al. paper). Since the protein encoded by Prg4, lubricin, has not been reported to be expressed in the intermediate/deep zone, it is likely that the Prg4 mRNA is not translated into proteins or that the protein is degraded. This is not that surprising because such a mode of step-wise shutting down of a gene, first at protein translational/posttranslational levels, and then at the transcriptional level has been reported in other systems during cell differentiation. For example, the neuronal specific repressor gene, Rest, is highly expressed in embryonic stem cells (ES), and when ES cells are committed to neural stem/progenitor cells, Rest protein are reduced remarkably, but Rest mRNA could still be detected. Only when neuronal terminal differentiation was accomplished, then Rest mRNA became no longer detectable (Ballas et al. *Cell*. 2005.121:645-657). It was also reported that superficial cells generate chondrocytes to reshape articular cartilage (Li et al. *FASEB Journal*. 2017.31.1067-1084). It is possible that the middle zonal chondrocyte while retain Prg4

gene transcription but protein expression was already shutting down. We added our explanation in newly revised manuscript (Line 253-256, Page10).

Figure Changes of Prg4 levels in each group in cluster C1, C2 and C3,.

Reviewers' Comments:

Reviewer #1:

Remarks to the Author:

In the revised manuscript, the authors answer the questions well.

Reviewer #2:

None

Reviewer #3:

Remarks to the Author:

Manuscript: NCOMMS-21-12281

The authors have done an excellent job addressing the points of concern from the previous review.

Authors adequately addressed the previous concern about how the 4 potential targets of miR-17 relevant to catabolism in OA cartilage were chosen in lines 172-176.

Authors addressed the previous minor concern about an explanation for why middle zonal chondrocytes express Prg4 mRNA but do not express the lubricin protein in lines 253-256.

The manuscript is ready to be accepted for publication.

REVIEWERS' COMMENTS

Reviewer #1 (Remarks to the Author):

In the revised manuscript, the authors answer the questions well.

Reviewer #3 (Remarks to the Author):

Manuscript: NCOMMS-21-12281

The authors have done an excellent job addressing the points of concern from the previous review.

Authors adequately addressed the previous concern about how the 4 potential targets of miR-17 relevant to catabolism in OA cartilage were chosen in lines 172-176.

Authors addressed the previous minor concern about an explanation for why middle zonal chondrocytes express Prg4 mRNA but do not express the lubricin protein in lines 253-256.

The manuscript is ready to be accepted for publication.

Response: Thanks for the reviewers' comments and helping us improve our manuscript.